# SELF-PACED AUGMENTATIONS (SPAUG) FOR IMPROVING MODEL ROBUSTNESS

## ABSTRACT

Augmentations are crucial components in modern computer vision. While various augmentation techniques have been devised to enhance model generalization and robustness, they are conventionally applied uniformly to all dataset samples during training. In this paper, we introduce "Self-Paced Augmentations (SPAug)," a novel approach that dynamically adjusts the augmentation intensity for each sample based on its training statistics. Our approach incurs little to no computational overhead and can be effortlessly integrated with existing augmentation policies with just a few lines of code. We integrate our self-paced augmentations into established uniform augmentation policies such as AugMix, RandomAugment, and AutoAugment. Our experiments reveal sizeable improvements, with about 1% enhancement on CIFAR10-C and CIFAR100-C datasets and a 1.81% improvement on ImageNet-C over AugMix, all while maintaining the same natural accuracy. Furthermore, within the context of augmentations designed to enhance model generalization, we demonstrate a 0.4% improvement over AutoAugment on CIFAR100, coupled with a 0.7% enhancement in model robustness.

## 1 INTRODUCTION

Data augmentations play a pivotal role in training Deep Neural Networks (DNNs) Shorten & Khoshgoftaar (2019); Perez & Wang (2017). They serve a dual purpose: firstly, as a regularization mechanism to counteract the tendency of models to overfit to the training dataset, and secondly, as a means to boost the model's robustness and generalization capability to previously unseen data Rebuffi et al. (2021); Van Dyk & Meng (2001); Wang et al. (2017). Considerable research efforts has been dedicated to finding optimal augmentation strategies, encompassing specific augmentation types and their associated magnitudes of application. These strategies result in improved model performance and robustness when applied during training. Prominent augmentation techniques such as AugMix Hendrycks et al. (2019), MixUp Zhang et al. (2017), CutMix Yun et al. (2019), CutOut DeVries & Taylor (2017), AutoAugment (AA) Cubuk et al. (2018), and RandomAugment (RA) Cubuk et al. (2020) have been introduced as mechanisms to achieve these objectives.

The augmentation parameters, comprising different augmentation types and their intensity of application, are typically determined based on a specific dataset in order to achieve a certain objective Shorten & Khoshgoftaar (2019). Additionally, over the course of training, these augmentation parameters remain *uniform* and are usually not tailored for individual data instances Müller & Hutter (2021); Zhang et al. (2018); Cubuk et al. (2018); Hendrycks et al. (2019); Cubuk et al. (2020). We argue that employing the same set of augmentation parameters for all data instances may not be optimal, given the diversity inherent in the dataset. For instance, certain samples, referred to as "*easy-samples*," rapidly converge to a lower loss (see blue curve in Figure 1), thereby allowing considerable room for synthetic augmentations of higher intensity. Conversely, other samples, denoted as "*hard-samples*," take longer time to fit during training – even in the absence of synthetic augmentations – which leaves minimal scope for incorporating synthetic augmentations (see green curve in Figure 1). As such, it is intuitive to consider adjusting the augmentation parameters to align with the training characteristics of each individual data instance over the course of training.

In this work, we explore the effectiveness of having *data instance-dependent augmentation parameters*, in contrast to conventional augmentation policies with uniform augmentation parameters for all data instances. One of our design choices is to enable instance-level augmentation intensities with *minimal computation overhead*. Solving all augmentation parameters per data instance cre-

ates a complex optimization problem. Hence, in this work, our focus is exclusively on adapting the augmentation intensity as the data instance parameter. During the training process, we regulate the intensity of augmentation infused into each sample by weighted-mixing of the original sample with its augmented counterpart. The data instance parameters controlling the augmentation intensity of each sample are updated during training, guided by the sample's training statistics. We employ the mixed sample's *training loss as a proxy measure* to gauge the augmentation intensity for the forthcoming epoch, thereby enabling the data instance parameter to fluctuate in accordance with training. We term this approach "*Self-Paced Augmentations*" (SPAug) because it dynamically adjusts the augmentation intensity based on the sample's convergence behavior.

Throughout our experiments, we demonstrate that SPAug not only accelerates convergence and enhances performance on clean test samples, it also exhibits noticeable robustness gains on common corruptions encountered during testing. SPAug introduces negligible computational overhead compared to training with uniform augmentation policies, and does not involve a complicated optimization phase. The improvements observed with corrupted test data highlight the efficacy of SPAug in enhancing model robustness. We integrate SPAug into a range of established fixed augmentation policies, including AugMix, (Hendrycks et al., 2019) AutoAugment (Cubuk et al., 2018), RandomAugment (Cubuk et al., 2020), and Adversarial Training (Madry et al., 2017) (See Appendix). In many instances, the addition of SPAug consistently leads to improved performance in terms of both clean and corrupted test error rates compared to the baseline, demonstrating its potential in being adapted as a common practice for training deep neural networks.

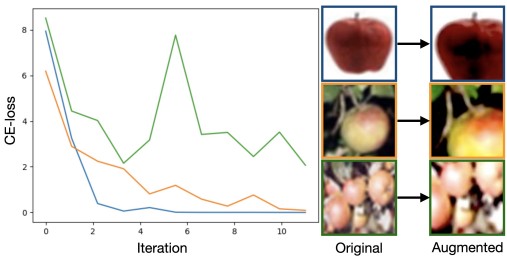

Figure 1: Convergence of training loss for an "easy", a "medium", and a "hard" sample. Easy samples converge rapidly and can accommodate higher levels of augmentation, while hard samples take longer to converge; applying the same level of augmentations as with easy samples can slower overall training convergence. We regulate the extent of augmentations applied to each sample based on its easiness. If the sample exhibits a low training loss, the intensity of augmentation will be increased in the subsequent epoch; conversely, if the loss is high, the intensity will be reduced.

## 2 RELATED WORK

### 2.1 UNIFORM AUGMENTATION POLICIES

Data augmentation policies involve applying a sequence of augmentations to a given sample during training. These augmentations can either be handcrafted or automatically optimized. *Handcrafted augmentations*, like AugMix (Hendrycks et al., 2019), are manually designed and inspired, with their hyperparameters tuned using the validation set. Conversely, *automatic augmentations*, such as AutoAugment (Cubuk et al., 2018), learn to dynamically determine the probabilities and magnitudes of applying different augmentation operations as a policy through reinforcement learning (RL) (Wiering & Van Otterlo, 2012; Kaelbling et al., 1996). The AutoAugment policy generator is updated by training a child model, with the validation accuracy serving as the reward. However, the search phase is computationally demanding. To address the large computational overhead, alternatives like Fast AutoAugment (Lim et al., 2019) rely on Bayesian optimization. In contrast, population-based training (PBT) (Ho et al., 2019) concurrently trains multiple child models in parallel, employing an evolutionary approach to discover the optimal augmentation policy. To streamline the search space, RandomAugment (Cubuk et al., 2020) suggests uniformly applying augmentation operations. RandomAugment involves parameterizing the search space solely with the count of augmentation operations and a global augmentation intensity. As opposed to using a validation set for evaluating augmentation policy quality, studies such as Adversarial AutoAugment (Zhang et al., 2019) employ adversarial objectives for learning the augmentation policy. *All these aforementioned augmentation policies treat all the samples in a dataset uniformly.* In contrast, we demonstrate that adjusting these uniform augmentation policies based on a sample's training statistics, such as its individual loss, can yield further performance enhancements.

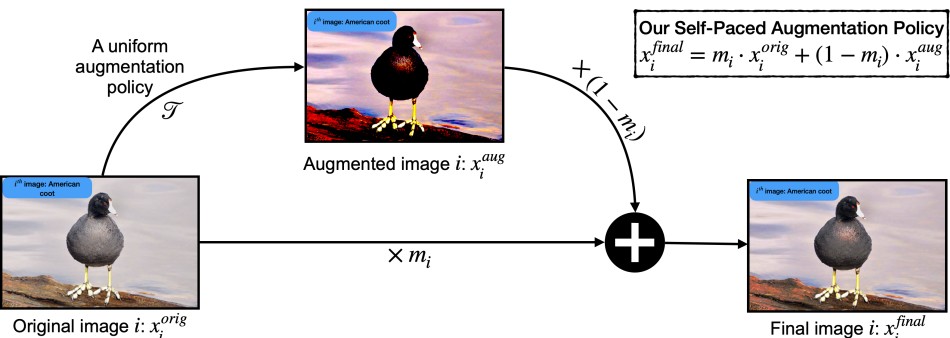

Figure 2: The formulation of our Self-Paced Augmentation (SPAug) policy. The parameters $m_i$ can be determined using hand-crafted functions like binary mapping function or polynomial mapping function, or it can be learned automatically during training alongside model parameters.

## 2.2 ADAPTIVE AUGMENTATION POLICIES

While uniform augmentation policies are most common, various efforts have been made to tailor augmentation policies to specific classes or subgroups. Hauberg et al. (2016) suggests adapting augmentation policies on a per-class basis by learning class-specific probabilistic generative models for transformations within a Riemannian submanifold (Chen, 2000) of the Lie group of diffeomorphisms. These learned augmentation policies, however, are confined to spatial transformations and deformations. Moreover, they assume that data must be locatable and alignable, rendering them less suitable for natural image datasets like CIFAR and ImageNet. A subsequent work called CAMEL (Goel et al., 2020) adopts a data-generation approach by addressing classifiers that struggle with a subgroup within a class. It trains a classifier using data augmentations intentionally designed to manipulate subgroup features. This is accomplished by employing a CycleGAN (Zhu et al., 2017) to learn intra-class, inter-subgroup augmentations. However, CAMEL necessitates manual specification of subgroup information and assumes that subgroups exist exclusively within the same class. MetaAugment (Zhou et al., 2021) introduces a meta-learning approach for adaptively adjusting augmented image loss weights on a per-sample basis and a global probability parameter that controls augmentation sampling. The task network is optimized by minimizing a weighted training loss, while the policy network aims to enhance the task network's performance on a validation set by tuning loss weights. In a more recent work, AdaAug (Cheung & Yeung, 2021) proposes class-dependent adaptive augmentation policies achieved through a policy network. The policy projection network is updated to minimize validation loss. However, the notably expansive search space – expanding with the number of training data samples – and the computationally demanding optimization of the policy network pose challenges for implementation on larger datasets. In contrast to all the above works, our method employs the augmented samples loss at the current epoch as a *surrogate measure* for adjusting augmentation intensity in its subsequent iteration. Our approach incurs little to no computational overhead, and can be effortlessly integrated with existing uniform augmentation policies, such as AugMix (Hendrycks et al., 2019), AutoAugment (Cubuk et al., 2018), RandomAugment (Cubuk et al., 2020), and Adversarial Training (Zhang et al., 2019; Shafahi et al., 2019), using just a few lines of code.

## 3 METHOD

### 3.1 SELF-PACED AUGMENTATION (SPAUG) FORMULATION

The conventional practice of applying uniform augmentation policies indiscriminately to all samples in a dataset often fails to account for the inherent disparities in the convergence rates of individual examples. While some samples readily converge during training, others require extended iterations to reach convergence (see Figure 1). The indiscriminate application of uniform augmentations can impede the efficiency of model training, particularly for challenging samples.

To address these challenges, we propose a dynamic augmentation strategy rooted in the concept of self-pacing. Our approach aims to adaptively adjust the intensity of augmentations for each sample during training, as illustrated in Figure 2. In this formulation, we assume a base uniform augmentation policy denoted as $\mathscr{T}$. For each sample $i$ in the training set, we introduce a single

parameter $m_i \in [0, 1]$ that controls the blending of the original (unaugmented) input $x_i^{orig}$ and the augmented input $x_i^{aug} = \mathscr{T}(x_i^{orig})$. This blending process yields the final input for model training, $x_i$, and is expressed as:

$$x_i = m_i \cdot x_i^{orig} + (1 - m_i) \cdot x_i^{aug}, \tag{1}$$

The conventional training approach, which applies a uniform policy $\mathscr{T}$ (e.g., AugMix, AutoAugment, etc.), can be regarded as a special case where $m_i = 0$ for all instances in the training set.

The key innovation of our method lies in the adaptive nature of $m_i$. Specifically:

- *For samples displaying a low training loss*, indicating ease of convergence, we increase the augmentation intensity in the subsequent training iteration,

- Conversely, *for samples with a high loss*, suggesting slower convergence, we decrease the augmentation intensity for that sample in the next iteration.

This adaptive strategy optimizes training convergence for both easy and hard samples, ensuring that each example receives a customized level of augmentation tailored to its unique characteristics. Sample easiness is determined by various metrics, including loss value (Bengio et al., 2009), entropy (Graves & Schmidhuber, 2008), distance from the decision boundary (Hacohen & Weinshall, 2019), and human expert annotations (Nili et al., 2014). In this work, we employ the primary loss value, typically Cross-Entropy (CE) (De Boer et al., 2005) for supervised classification, as the simplest proxy for easiness.

## 3.2 HAND-CRAFTED FUNCTIONS FOR PER-INSTANCE PARAMETER ESTIMATION

To implement the concept of self-pacing, we need functions that map sample easiness (i.e., low loss values) to values of $m_i$. As aforementioned, these functions are crafted to augment easy examples more intensely than hard ones, thus promoting balanced convergence during training. We draw inspiration from curriculum learning literature to formulate these functions (Bengio et al., 2009; Wang et al., 2021; Jiang et al., 2014).

**Binary Mapping Functions.** One of the simplest mapping functions is binary, which assigns $m_i = 1$ if sample $i$ is considered hard and $m_i = 0$ if it is regarded as easy. This binary mapping function can be expressed mathematically as (Kumar et al., 2010):

$$m_i = H(\mathcal{L}_i) = \begin{cases} 1, & \text{if } \mathcal{L}_i \geq \tau \\ 0, & \text{otherwise,} \end{cases} \tag{2}$$

In this context, the hyperparameter $\tau$ acts as the threshold distinguishing easy from hard samples. The variable $\mathcal{L}_i$ represents the primary loss of $x_i$, specifically the CE loss for classification. The pseudocode for SPAug with binary mapping function is given in Algorithm 1. As demonstrated, SPAug with binary mapping function can be added to the training code only using 5 lines of code.

**Continuous Mapping Functions.** In a more flexible scenario, per-instance augmentation parameters may take non-binary values, allowing for a fine-grained adjustments. We utilize polynomial mapping functions for this purpose (Gong et al., 2018):

$$m_i = P(\mathcal{L}_i) = \begin{cases} 1 - (1 - \frac{\mathcal{L}_i}{\tau})^{\frac{1}{t-1}}, & \text{if } \mathcal{L}_i \geq \tau \\ 0, & \text{otherwise,} \end{cases} \tag{3}$$

Here, $\tau$ represents the threshold for distinguishing easy samples from hard ones, while $t$ governs the shape of the polynomial mapping function.

## 3.3 LEARNING PER-INSTANCE PARAMETERS

We extend our SPAug by optimizing the augmentation parameters $m_i$ alongside model parameters. We achieve this objective with the following loss function:

$$L = \mathcal{L}_i - \sigma(m_i) \cdot \text{Sign}\left(\mathcal{L}_i - [\mathcal{L}_{\min} + \tau \cdot (\mathcal{L}_{\max} - \mathcal{L}_{\min})]\right) \tag{4}$$

Here, $\mathcal{L}_i$ represents the primary loss (CE loss) of sample $i$, $\sigma(\cdot)$ denotes the Sigmoid function, $\mathcal{L}_{\min}$ and $\mathcal{L}_{\max}$ are the minimum and maximum loss values within the

$$\mathscr{L}_\tau = \mathscr{L}_{min} + \tau \cdot (\mathscr{L}_{max} - \mathscr{L}_{min})$$

*Easy samples* ←$\mathscr{L}_\tau$→ *Hard samples*

Figure 3: Understanding the regularization loss in learnable SPAug.

batch, and $\tau$ is the threshold that distinguishes easy samples from hard ones in the minibatch. The second term in Equation (4) is the regularization term which we parameterize as $\mathcal{L}_r$. To ensure the final blending weight for $x_i^{\text{orig}}$ and $x_i^{\text{aug}}$ remains between 0 and 1, we apply the Sigmoid activation to the instance parameter $m_i$. Consequently, Equation (1) is adopted for learnable SPAug as: $x_i = \sigma(m_i) \cdot x_i^{\text{orig}} + (1 - \sigma(m_i)) \cdot x_i^{\text{aug}}$. During optimization of Equation (4), model weights receive updates based on gradients from the primary loss $\mathcal{L}_i$, while the instance parameters are updated using both loss terms: $\mathcal{L}_i$ and $\mathcal{L}_r$. As illustrated in Figure 3.3, Equation (4) can be divided into two scenarios. When a sample $i$ is considered as easy (i.e., $\mathcal{L}_i$ is less than the threshold loss $\mathcal{L}_\tau$ – determined based on the batch's loss values), we minimize $L = \mathcal{L}_i + \sigma(m_i)$. In this scenario, the regularization term guides $m_i$'s towards lower values, preventing $m_i$'s to converge to degenerate solutions where $m_i = \infty$. Conversely, for hard samples, we minimize $L = \mathcal{L}_i - \sigma(m_i)$, resulting in lower levels of augmentations in the subsequent epoch. It's important to note that the classification of samples as easy or hard is determined by comparing the loss values within the current minibatch. Over the course of training, the state (i.e., easiness/hardness) of these samples evolve based on how quickly they converge relative to other samples and the statistics of the minibatch. To optimize data-instance parameters and model parameters separately, we utilize two distinct optimizers. Since the number of instance parameters to optimize in a given batch is relatively small (equal to the batch size), this introduces minimal computational overhead compared to standard training.

**Algorithm 1** Self-Paced Augmentation w/ Binary Mapping Functions' Pseudo-Code

```
1    m = torch.ones(len_dataset) # SPAug
2    tau = 0.5 # SPAug
3
4    for epoch in range(epochs):
5        for i, x, y in loader:
6            mi = (m[i] > tau).float().view((-1, 1, 1, 1)) # SPAug
7            x_final = mi * x + (1 - mi) * aug(x) # SPAug
8
9            main_loss = loss(model(x_final), y)
10            main_loss.mean().backward()
11            optimizer.step()
12            optimizer.zero_grad()
13
14            m[i] = main_loss.clone().detach() # SPAug
15
```

## 4 EXPERIMENTS

### 4.1 EXPERIMENTAL SETUP

In our experiments, we focus on CIFAR (Krizhevsky, 2009) and ImageNet1K (Deng et al., 2009) datasets and assess the impact of our SPAug method in conjunction with established uniform augmentation policies. Specifically, we employ the following uniform augmentation strategies $\mathscr{T}$: Aug-Mix (Hendrycks & Dietterich, 2019), AutoAugment (Cubuk et al., 2018), RandomAugment (Cubuk et al., 2020), and Adversarial Training[1]. These augmentation policies have been proven effective in enhancing model robustness and generalization. For evaluating model performance, we consider both clean test data and corrupted test sets (CIFAR10-C, CIFAR100-C, and ImageNet-C) (Hendrycks & Dietterich, 2019). It's worth noting that AugMix training results are reported under two scenarios: training solely with cross-entropy loss and training with an additional regularizer called Jenson-Shannon-Divergence loss (JSD). Hence, our experiments with AugMix encompass a comprehensive comparison of these scenarios. In the case of AutoAugment (Cubuk et al., 2018), we apply our self-paced augmentations alongside the optimal augmentation policy designed for CIFAR datasets. Additionally, we investigate scenarios where a mismatched augmentation policy is used, a practical choice at times due to the extensive search process required by AutoAugment. For RandomAugment we use parameters $N = 3$ and $M = 4$ when training with CIFAR datasets, and we incorporate SPAug into the training pipeline. For CIFAR experiments, our main network architecture is based on WideResNets (WRN) Zagoruyko & Komodakis (2016).[2] Our choice of architecture

---

[1]Proof-of-concept experiments for adversarial training can be found in the supplementary materials.

[2]We include experiments with other architecture choices in the Appendix.

for ImageNet experiments is ResNet50 (He et al., 2016). For CIFAR experiments, all networks are trained with an initial learning rate of 0.1, which undergoes decay following a cosine learning rate schedule (Loshchilov & Hutter, 2016). We apply standard preprocessing to input images, including random horizontal flipping and cropping, before any augmentations are introduced (denoted as original image $x^{org}$). Training is carried out using SGD with Nesterov momentum (Sutskever et al., 2013) and use weight decay of 0.0005 (Zhang et al., 2017; Guo et al., 2019). In cases where learnable instance parameters are used within the SPAug framework, we employ the AdamW optimizer (Kingma & Ba, 2014) with a learning rate of 0.01 and a weight decay of 0.0005 for optimizing $m_i$. All experiments are repeated three times, and average classification error and standard deviation are reported.

## 4.2 A TOY EXPERIMENT

Table 1: Results from the CIFAR100 toy experiment highlighting the significance of SPAug-Binary. Applying $\mathcal{T} = \mathcal{T}_h$ to all samples ($\tau = \infty$) or to no samples ($\tau = 0$) during training yields models with higher clean and corrupted (abbreviated as corr.) errors compared to those using SPAug-Binary.

| Epochs | Err. | Threshold - $\tau$ | | | | | | |
|---|---|---|---|---|---|---|---|---|
| | | 0 | 0.1 | 0.2 | 0.4 | 0.6 | 0.8 | $\infty$ |
| 50 | clean | $25.8_{\pm.4}$ | $25.8_{\pm.5}$ | $\mathbf{25.3}_{\pm.2}$ | $26.0_{\pm.2}$ | $26.3_{\pm.4}$ | $26.5_{\pm.2}$ | $32.7_{\pm.2}$ |
| | corr. | $54.9_{\pm.3}$ | $\mathbf{51.4}_{\pm.4}$ | $51.5_{\pm.2}$ | $52.0_{\pm.2}$ | $52.1_{\pm.2}$ | $52.6_{\pm.4}$ | $56.6_{\pm.6}$ |
| 100 | clean | $24.4_{\pm.2}$ | $\mathbf{24.0}_{\pm.1}$ | $24.2_{\pm.1}$ | $24.6_{\pm.2}$ | $24.4_{\pm.5}$ | $24.9_{\pm.2}$ | $29.6_{\pm.4}$ |
| | corr. | $53.6_{\pm.3}$ | $51.0_{\pm.2}$ | $\mathbf{50.9}_{\pm.3}$ | $51.1_{\pm.1}$ | $51.5_{\pm.2}$ | $51.3_{\pm.3}$ | $54.3_{\pm.2}$ |

We begin our analysis with a simple toy experiment to demonstrate the importance of having data-instance augmentation parameters when training a DNN. We train a WideResNet-40-2 on CIFAR-100 with a hand-crated augmentation policy $\mathcal{T}_h$, which includes CIFAR100 basic augmentations followed by random rotation and random color transformations. We use simple binary mapping function (parameterized by threshold $\tau$) to determine the data instance parameters ($m_i \in \{0, 1\}$). Results are summarized in Table 1.

In Table 1, $\tau = 0$ corresponds to training with only basic augmentations, while $\tau = \infty$ corresponds to training with uniform augmentation policy. When, $0 < \tau < \infty$, only samples with CE-loss below the threshold $\tau$ in the previous iteration, undergo $\mathcal{T}_h$ in the subsequent iteration; otherwise, they train with basic augmentations. In Table 1 we observe that the lowest clean and corrupted test error is observed for threshold of $0.1$ or $0.2$, which implies that frequently adding augmentation to easy samples (samples with low loss) while rarely adding augmentations to hard samples results in a robust model which perform well on both natural and corrupted data. Noticeably, we observe, considerable improvement in corruption error (e.g., **up to 3.5% in absolute terms**), preliminary demonstrating the potential for improving the model robustness while not sacrificing clean accuracy.

To gain further insight into how the binary mapping function governs the extent of augmentations incorporated into a given sample during the training process, we present a visual representation in Figure 4. Easy samples, as shown in Figure 4-(a) and (b), experience augmented images more frequently. Moderate samples, shown in Figure 4-(c), require more time to converge than easy samples, resulting in training with augmented images only halfway in training. Conversely, hard samples, shown in Figure 4-(d), which inherently possess natural augmentations that make them challenging to fit, are rarely exposed to the synthetic augmentation policy.

## 4.3 RESULTS WITH AUGMIX POLICY

Building upon the encouraging results outlined in Section 4.2, our objective is to improve the performance of established uniform augmentation strategies. Our approach revolves around dynamically adjusting the extent of augmentation applied to each training sample, guided by their individual training loss. To kick off our investigation, we begin by analyzing the AugMix policy (Hendrycks et al., 2019), which was initially designed to enhance model robustness against corrupted data while preserving clean accuracy.

Hendrycks et al. (2019) present results under two scenarios: (1) training with CE loss, and (2) training with CE loss supplemented by an additional regularizer, known as Jenson-Shannon Divergence (JSD) loss (Menéndez et al., 1997). To maintain consistency, we report results for both of these setups in Table 2. The table summarizes outcomes for various configurations, including the baseline,

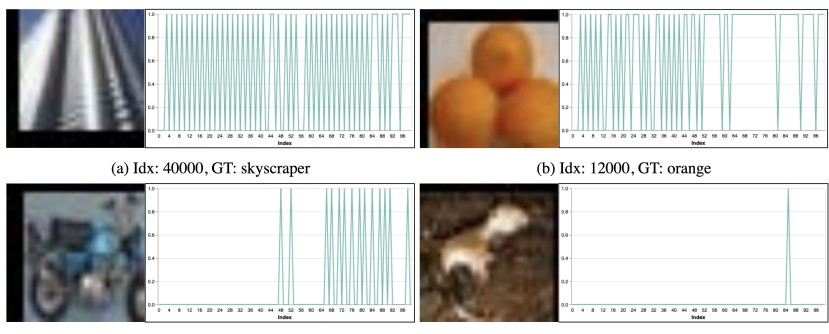

(a) Idx: 40000, GT: skyscraper

(b) Idx: 12000, GT: orange

(c) Idx: 0, GT: bike

(d) Idx: 11000, GT: possum

Figure 4: How the binary mapping function governs the extent of augmentations applied to each sample during training for (a-b) easy, (c) moderate, and (d) hard samples.

Table 2: Comparison of corrupted error results for AugMix augmentation policy used as base policy with WRN-40-2 and WRN-28-10.

| Dataset | JSD ? | Baseline | AugMix | SPAug-Binary | SPAug-Poly | SPAug-Learnable |
|---------|-------|----------|--------|--------------|------------|-----------------|
| **WRN-40-2** | | | | | | |
| Cifar-10 | ✗ | $27.2_{\pm.3}$ | $14.0_{\pm.4}$ | $13.2_{\pm.1}$ | $13.2_{\pm.1}$ | $\mathbf{13.0_{\pm.2}}$ |
| | ✓ | | $11.2_{\pm.3}$ | $11.5_{\pm.2}$ | $11.0_{\pm.2}$ | $\mathbf{11.0_{\pm.1}}$ |
| Cifar-100 | ✗ | $53.2_{\pm.3}$ | $40.0_{\pm.1}$ | $39.5_{\pm.2}$ | $39.1_{\pm.3}$ | $\mathbf{39.0_{\pm.1}}$ |
| | ✓ | | $36.1_{\pm.1}$ | $35.4_{\pm.1}$ | $35.2_{\pm.4}$ | $\mathbf{35.0_{\pm.1}}$ |
| **WRN-28-10** | | | | | | |
| Cifar-10 | ✗ | $23.3_{\pm.3}$ | $10.7_{\pm.2}$ | $10.2_{\pm.1}$ | $10.1_{\pm.2}$ | $\mathbf{10.0_{\pm.1}}$ |
| | ✓ | | $9.3_{\pm.0}$ | $9.1_{\pm.1}$ | $8.8_{\pm.2}$ | $\mathbf{8.7_{\pm.0}}$ |
| Cifar-100 | ✗ | $48.0_{\pm.2}$ | $33.6_{\pm.3}$ | $33.4_{\pm.3}$ | $33.3_{\pm.2}$ | $\mathbf{33.3_{\pm.0}}$ |
| | ✓ | | $31.9_{\pm.2}$ | $31.7_{\pm.1}$ | $31.5_{\pm.2}$ | $\mathbf{31.3_{\pm.1}}$ |

default AugMix, AugMix with Binary mapping SPAug, AugMix with polynomial mapping SPAug, and AugMix with learnable SPAug. Since AugMix primarily targets reduction of corrupted errors, we emphasize this metric in Table 2 (for clean error rates, see the supplementary material.)

The observed results affirm that integrating AugMix augmentation significantly enhances performance on the corrupted test set. Note that the adaptive adjustment of augmentation intensity, based on sample loss using binary mapping and polynomial panning functions, improves model robustness against corrupted test data. The polynomial mapping function outperforms the binary mapping function in terms of reduced corrupted test error. This outcome is intuitive, as the polynomial mapping function enables a spectrum of augmentation intensities based on loss unlike the binary mapping function. In addition, learning instance intensity parameters, rather than relying on handcrafted mapping functions, leads to enhanced performance, particularly evident in the corrupted test set. Thus, the results from the AugMix augmentation policy suggest that incorporating sample-dependent intensity parameters based on loss evaluation can improve model performance. As learnable instance parameters demonstrate effectiveness over hand-crafted mapping functions, we adopt it as our default setting for subsequent experiments.

To demonstrate the applicability of our approach on larger datasets, we perform similar experiments using AugMix on ImageNet. We use a ResNet50 architecture and train for 270 epochs. We use the AugMix without the JSD regularization. The results are summarized in Table 3.

Table 3: Clean Error (Err.) and Corrupted Error (C-Err.) for ResNet50 ImageNet models trained with AugMix policy.

| | Err. | C-Err. |
|---|------|--------|
| AugMix | $\mathbf{23.1_{\pm.1}}$ | $74.35_{\pm.1}$ |
| SPAug-Learnable | $\mathbf{23.1_{\pm.1}}$ | $\mathbf{72.54_{\pm.1}}$ |

## 4.4 RESULTS WITH AUTOAUGMENT

In this section, we conduct experiments with the AutoAugment augmentation policy, a widely adopted uniform augmentation strategy. As discussed in the related work section, AutoAugment determines the optimal augmentation policy via reinforcement learning, maximizing performance on the validation set. We employ the default AutoAugment policy for CIFAR and introduce our learn-

Table 4: Comparison of results for AutoAugment Cubuk et al. (2018) with WRN-28-10.

| Dataset | # epochs | Baseline | | AA | | SPAug-Learnable | |
|---|---|---|---|---|---|---|---|
| | | Err. | C-Err. | Err. | C-Err. | Err. | C-Err. |
| **CIFAR datasets with AutoAugment CIFAR policy** | | | | | | | |
| CIFAR-10 | 100 | $3.9_{\pm.1}$ | $24.1_{\pm.1}$ | $\textbf{3.3}_{\pm.2}$ | $16.2_{\pm.2}$ | $\textbf{3.3}_{\pm.1}$ | $\textbf{14.8}_{\pm.2}$ |
| | 200 | $3.8_{\pm.0}$ | $23.5_{\pm.9}$ | $3.0_{\pm.2}$ | $15.6_{\pm.3}$ | $\textbf{2.9}_{\pm.1}$ | $\textbf{14.3}_{\pm.3}$ |
| CIFAR-100 | 100 | $19.2_{\pm.1}$ | $48.3_{\pm.2}$ | $18.1_{\pm.2}$ | $40.5_{\pm.1}$ | $\textbf{17.9}_{\pm.1}$ | $\textbf{39.3}_{\pm.2}$ |
| | 200 | $18.5_{\pm.3}$ | $47.5_{\pm.3}$ | $17.6_{\pm.1}$ | $39.6_{\pm.0}$ | $\textbf{17.2}_{\pm.2}$ | $\textbf{38.9}_{\pm.1}$ |
| **CIFAR datasets with AutoAugment ImageNet policy** | | | | | | | |
| CIFAR-10 | 100 | $3.9_{\pm.1}$ | $24.1_{\pm.1}$ | $3.8_{\pm.0}$ | $14.2_{\pm.3}$ | $\textbf{3.6}_{\pm.1}$ | $\textbf{12.1}_{\pm.2}$ |
| | 200 | $3.8_{\pm.0}$ | $23.5_{\pm.9}$ | $3.5_{\pm.0}$ | $13.1_{\pm.0}$ | $\textbf{3.4}_{\pm.0}$ | $\textbf{12.0}_{\pm.2}$ |
| CIFAR-100 | 100 | $19.2_{\pm.1}$ | $48.3_{\pm.2}$ | $19.5_{\pm.6}$ | $37.3_{\pm.4}$ | $\textbf{19.0}_{\pm.3}$ | $\textbf{35.7}_{\pm.2}$ |
| | 200 | $\textbf{18.5}_{\pm.3}$ | $47.5_{\pm.3}$ | $19.2_{\pm.2}$ | $36.4_{\pm.4}$ | $18.8_{\pm.5}$ | $\textbf{35.5}_{\pm.2}$ |
| **CIFAR datasets with AutoAugment SVHN policy** | | | | | | | |
| CIFAR-10 | 100 | $3.9_{\pm.1}$ | $24.1_{\pm.1}$ | $3.6_{\pm.1}$ | $16.3_{\pm.1}$ | $\textbf{3.5}_{\pm.1}$ | $\textbf{12.7}_{\pm.2}$ |
| | 200 | $3.8_{\pm.0}$ | $23.5_{\pm.9}$ | $3.4_{\pm.1}$ | $15.9_{\pm.1}$ | $\textbf{3.3}_{\pm.2}$ | $\textbf{13.0}_{\pm.0}$ |
| CIFAR-100 | 100 | $\textbf{19.2}_{\pm.1}$ | $48.3_{\pm.2}$ | $19.4_{\pm.2}$ | $42.0_{\pm.1}$ | $19.4_{\pm.3}$ | $\textbf{36.5}_{\pm.2}$ |
| | 200 | $\textbf{18.5}_{\pm.3}$ | $47.5_{\pm.3}$ | $\textbf{18.5}_{\pm.2}$ | $41.1_{\pm.6}$ | $18.6_{\pm.2}$ | $\textbf{37.8}_{\pm.3}$ |

able instance parameters to dynamically regulate the augmentation intensity for each sample during training, departing from uniform application. The results are summarized in Table 4, considering two scenarios: (1) training CIFAR with CIFAR policy, and (2) training CIFAR with mismatched AutoAugment policies optimized for ImageNet and SVHN datasets.

Let's first examine the case of training CIFAR datasets with their optimal policy. It becomes evident that incorporating SPAug-Learnable enhances the default AutoAugment outcomes for both the clean and corrupted test sets, for both CIFAR-10 and CIFAR-100, regardless of training for 100 or 200 epochs. Notably, the application of SPAug a significant improvement in the performance on the corrupted test set (denoted as C-Err.).

We also explore the implications of training CIFAR datasets with mismatched augmentation policies. Given the computationally intensive nature of the AutoAugment search process, a known mismatch policy might be adopted when training a model on a new dataset. This experiment emulates such behavior. Expectedly, we observe a decrease in model performance on natural test images when trained with a mismatched AutoAugment policy. However, in some instances, the performance on corrupted test images sees an improvement. Nonetheless, incorporating SPAug-Learnable to control the augmentation intensity for each sample further enhances performance for both natural and corrupted test sets. This showcases SPAugs applicability across a broad spectrum of scenarios.

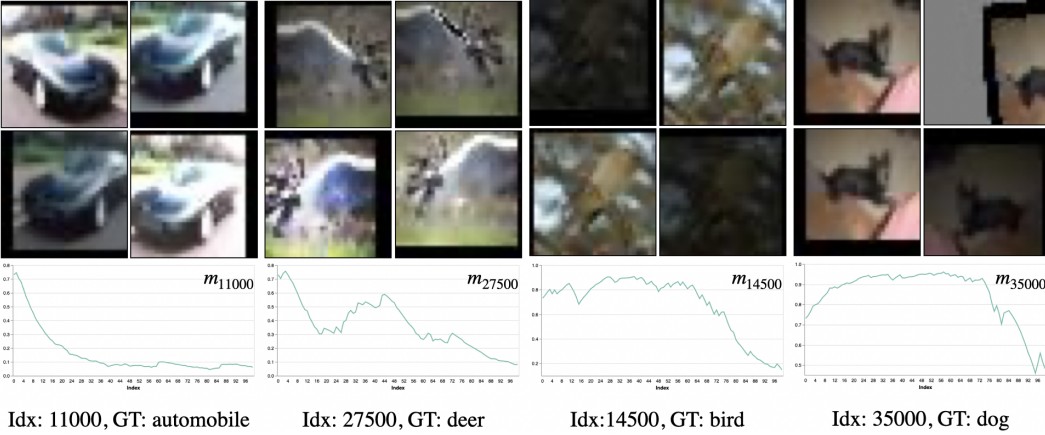

Idx: 11000, GT: automobile    Idx: 27500, GT: deer    Idx:14500, GT: bird    Idx: 35000, GT: dog

Figure 5: Visualization of how augmentation instance parameters ($m_i$) vary during training with AA augmentations for different sample types ("easy", "medium", and "hard") in CIFAR-10.

To illustrate the evolving nature of learnable instance parameters during the training process for various samples of the CIFAR-10 dataset, we provide visualizations in Figure 5. As depicted, instance parameter values for easy samples (automobile) consistently decline, nearing a value close to 0. This trend suggests a preference for training with heightened levels of AA augmentations. Conversely, certain samples like deer exhibit a gradual decline, followed by an increase that stabilizes at a moderately high value before ultimately dropping to 0 by the training's conclusion. Conversely, for samples like bird, and dog, characterized by inherent natural augmentations and greater complexity, converge very slowly. Consequently, training with AA augmentation of significant magnitude only occurs towards the end of the training process for those harder samples.

## 4.5 RESULTS WITH RANDOMAUGMENT

Table 5: Comparison of results for RandomAugment (Cubuk et al., 2020) augmentation policies with WRN-28-10. The reported results are the average of 3 random trials.

| Dataset | # epochs | RA | | SPAug-Learnable | |
|---|---|---|---|---|---|
| | | clean | corr. | clean | corr. |
| Cifar-10 | 100 | $3.7_{\pm.1}$ | $14.4_{\pm.3}$ | $3.8_{\pm.1}$ | $12.5_{\pm.0}$ |
| | 200 | $3.3_{\pm.1}$ | $13.6_{\pm.2}$ | $3.5_{\pm.1}$ | $13.3_{\pm.2}$ |
| Cifar-100 | 100 | $19.6_{\pm.1}$ | $41.4_{\pm.3}$ | $19.4_{\pm.1}$ | $38.0_{\pm.2}$ |
| | 200 | $19.1_{\pm.2}$ | $40.3_{\pm.1}$ | $19.1_{\pm.1}$ | $38.4_{\pm.1}$ |

In this section, we conduct experiments involving the RandomAugment augmentation policy. RandomAugment was introduced as a means to circumvent the costly and expansive search space associated with AutoAugment. RandomAugment operates by adopting two augmentation parameters: the count of applied augmentations and a global magnitude parameter for each augmentation. We incorporate our adaptive data instance augmentation parameters onto the RandomAugment strategy to regulate the intensity of application for each sample. The outcomes are summarized in Table 4.

Similar to our observations with other augmentation policies, we note an enhancement in performance across both natural and corrupted test sets when employing learnable instance parameters. While RandomAugment occasionally yields superior results in natural accuracy, it's particularly noteworthy that we witness a substantial boost in corrupted test set performance when integrating learnable instance parameters. This highlights its efficacy in fortifying the model against diverse forms of corruption.

## 5 CONCLUSION

In this paper, we introduced self-paced augmentations (SPAug), a technique for controlling the augmentation intensity experienced by individual training samples based on their training statistics. Our approach diverges from complex and computationally expensive inner loop optimization that typically relies on a validation set to determine instance-specific augmentation parameters. Instead, we employ the sample's training loss as a proxy measure to ascertain the level of augmentations in the subsequent iteration. We applied our self-paced augmentations on top of existing uniform augmentation policies, including AugMix, AutoAugment, RandomAugment, and Adversarial Training. The results demonstrate performance improvements, both in terms of natural test accuracy and corrupted test accuracy. Notably, we observed significant enhancements in corrupted test accuracy, showcasing our method's effectiveness in enhancing model robustness against unseen corruptions while preserving natural accuracy. One of the key advantages of adopting any of the SPAug variations is their ease of integration into any existing uniform augmentation policy, requiring just a few lines of code adjustments. Computationally, SPAug introduces minimal overhead, rendering it highly suitable for practical implementation in day-to-day deep learning model training.

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
