# SELF-PACED AUGMENTATIONS (SPAUG) FOR IMPROVING MODEL ROBUSTNESS

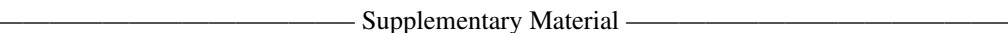
Supplementary Material

## A  RESULTS WITH ADVERSARIAL TRAINING

While deep neural networks have exhibited unparalleled performance across numerous domains, they have also unveiled a perplexing Achilles' heel – their susceptibility to adversarial examples (**??**). Adversarial examples are crafted with malicious intent, designed to deceive or disrupt machine learning models. To ensure these perturbed inputs appear nearly identical to their benign counterparts to human observers, adversarial attackers bound the adversarial noise to $\ell_p$-norm bounds (**?**). In such settings, the adversary perturbs the input by optimizing the following loss:

$$\max L_{main}(x_i^{adv}) = \max_{|\delta|_p \leq \epsilon} L_{main}(x_i^{clean} + \delta) \tag{1}$$

where $L_{main}$ is often the Cross-entropy loss, $\delta$ is the adversarial perturbation, and $\epsilon$ is the adversarial perturbation budget.

To this day, the defacto defense method against adversarial examples is adversarial training, which involves on the fly generation of adversarial examples and updating the network weights to become robust against adversarial examples (**???**). In this section, we focus on the work of **?** which achieves the highest performance by optimizing the parameters of adversarial training.

Considering that $x_i^{adv}$ as an augmented version of $x_i^{clean}$, we adopt self-paced augmentations (SPAug) for adversarial training. Adversarial training methods enforce a uniform $\ell_p$-norm bound for all training data instances. Adopting SPAug is equivalent to tailoring the $\ell_p$-norm bound for every instance during adversarial training.

$$x_i^{final} = \sigma(m_i) \cdot x_i^{clean} + (1 - \sigma(m_i)) \cdot x_i^{adv}$$

For the experimental setup, we use the setting described in **?** for ResNet18 architecture on CIFAR-100 dataset. Table 1 summarizes the results of **?** with and without self-paced augmentations. As it can be seen, incorperating SPAug into adversarial training results in reduction of all three errors, clean error (E), corrupter error (C-E), and adversarial error (R-E). The adversarial error is computed by running AutoAttack (AA) (**?**) on the clean test data. This preliminary result further re-enforces the applicability of SPAug to various choices of base augmentation policies.

Table 1: Adversarial Augmentations (i.e., adversarial training) results with RN18 backbone. Numbers reported are errors (lower is better).

| Dataset | # epochs | Adv | | | SPAug-Learnable (ours) | | |
|---|---|---|---|---|---|---|---|
| | | E | C-E | AA(R-E) | E | C-E | AA(R-E) |
| CIFAR-100 | 110 | $42.06_{\pm.79}$ | $52.35_{\pm.49}$ | $76.24_{\pm.19}$ | $\mathbf{41.54_{\pm 1.18}}$ | $\mathbf{51.86_{\pm 1.03}}$ | $\mathbf{76.00_{\pm.22}}$ |

## B AUGMIX RESULTS ON VARIOUS CHOICES OF ARCHITECTURES

In this section, we present a comparative analysis of the Augmix data augmentation technique both with and without the incorporation of the learnable SPAug module on the CIFAR-10 and CIFAR-100 datasets. To further underscore the efficacy and versatility of SPAug, we conducted experiments that mirror the setup presented in Table 2, utilizing various architectural models, including WideResNets (**?**), AllConvNet (**?**), and DenseNet (**?**). Notably, our findings indicate that the inclusion of SPAug in conjunction with Augmix consistently results in improved error rates across all the examined network architectures.

Table 2: **Comparing AugMix and Learnable-AugMix across multiple networks. The reported results are the average of 3 random trials.**

| Dataset | Network | JSD | AugMix | | AugMix-Learnable | |
|---|---|---|---|---|---|---|
| | | | clean | corr. | clean | corr. |
| Cifar-10 | AllConvNet | ✗ | **6.2** | 18.4 | 6.3 | **16.7** |
| | DenseNet | ✗ | **4.9** | 14.8 | 5.0 | **14.0** |
| | | ✓ | 4.6 | 12.0 | 4.6 | **11.5** |
| | WRN-40-2 | ✗ | **4.9** | 14.0 | 5.0 | **13.0** |
| | | ✓ | 4.7 | 11.2 | **4.6** | **11.0** |
| | WRN-28-10 | ✗ | 3.8 | 10.7 | **3.6** | **10.0** |
| | | ✓ | 3.6 | 9.3 | **3.5** | **8.7** |
| Cifar-100 | AllConvNet | ✗ | **6.2** | 18.4 | 6.3 | **16.7** |
| | DenseNet | ✗ | **4.9** | 14.8 | 5.0 | **14.0** |
| | | ✓ | 4.6 | 12.0 | 4.6 | **11.5** |
| | WRN-40-2 | ✗ | **4.9** | 14.0 | 5.0 | **13.0** |
| | | ✓ | 4.7 | 11.2 | **4.6** | **11.0** |
| | WRN-28-10 | ✗ | 3.8 | 10.7 | **3.6** | **10.0** |
| | | ✓ | 3.6 | 9.3 | **3.5** | **8.7** |

## C  HYPERPARAMETER TUNING

Table 3: Hyperparamter tuning results in CIFAR-10 dataset with learnable-SPAug with AugMix policy.

| Architecture | AugMix | | Learnable-SPAug with varying $\tau$ | | | | | | | | | | | | |
| --- | --- | --- | --- | --- | --- | --- | --- | --- | --- | --- | --- | --- | --- | --- | --- |
| | | | 0.0 | | 0.1 | | 0.25 | | 0.5 | | 0.75 | | 0.9 | | 1.0 | |
| | E | C-E | E | C-E | E | C-E | E | C-E | E | C-E | E | C-E | E | C-E | E | C-E |
| WRN-40-2 | 4.9 | 14.0 | 5.2 | 27.2 | 5.1 | 13.5 | 5.1 | 13.1 | 5.2 | 13.1 | 5.0 | **13.1** | **5.0** | **13.0** | 5.0 | 13.1 |
| WRN-28-10 | 3.8 | 10.7 | 3.9 | 25.6 | 3.6 | 10.3 | 3.7 | 10.1 | **3.6** | **10.0** | 3.6 | 10.2 | 3.7 | 10.1 | 3.7 | 10.1 |

Table 4: Hyperparamter tuning results in CIFAR-100 dataset with learnable-SPAug with AugMix policy.

| Architecture | AugMix | | Learnable-SPAug with varying $\tau$ | | | | | | | | | | | | |
| --- | --- | --- | --- | --- | --- | --- | --- | --- | --- | --- | --- | --- | --- | --- | --- |
| | | | 0.0 | | 0.1 | | 0.25 | | 0.5 | | 0.75 | | 0.9 | | 1.0 | |
| | E | C-E | E | C-E | E | C-E | E | C-E | E | C-E | E | C-E | E | C-E | E | C-E |
| WRN-40-2 | 24.3 | 40.0 | 24.2 | 53.2 | 24.6 | 40.6 | 24.8 | 40.0 | 24.6 | 39.5 | **24.5** | **38.9** | 24.7 | 39.0 | 24.3 | 38.9 |
| WRN-28-10 | 19.5 | 33.6 | 19.2 | 48.0 | 20.0 | 34.0 | 20.2 | 33.9 | 20.1 | 33.3 | **20.0** | **33.3** | **20.1** | **33.5** | 20.1 | 33.4 |

SPAug requires the selection of a hyperparameter, $\tau$, which defines the boundary between easy and hard samples within a given batch. To identify the optimal hyperparameter value, we conduct a grid search using the following values: 0, 0.1, 0.25, 0.50, 0.75, 0.9, and 1.0 for learnable-SPAug. The results of this search are presented in Table 3 and 4.

## D  ADDITIONAL VISUALIZATIONS OF DATA-INSTANCE PARAMETERS

Figures 1-6 provides additional visualizations of how the data-instance parameter varied over the course of training for Learnable-SPAug experiments with AutoAugment augmentations on CIFAR10 (Figure 1, 2, 3) and CIFAR100 (Figure 4, 5, 6).

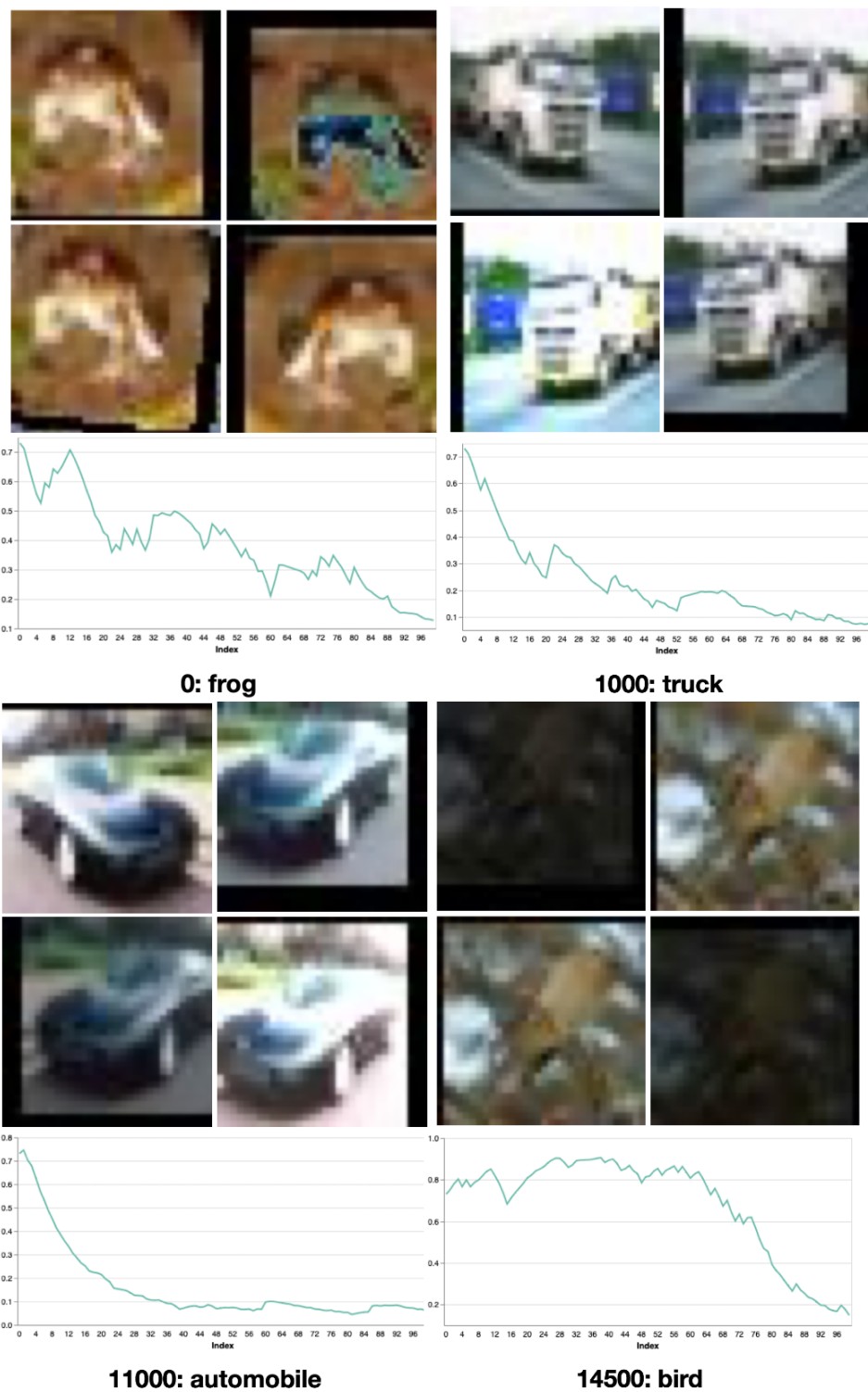

Figure 1: Additional visualizations of how augmentation instance parameters $(m_i)$ vary during training with AA augmentations for different sample types ("easy", "medium", and "hard") in CIFAR-10.

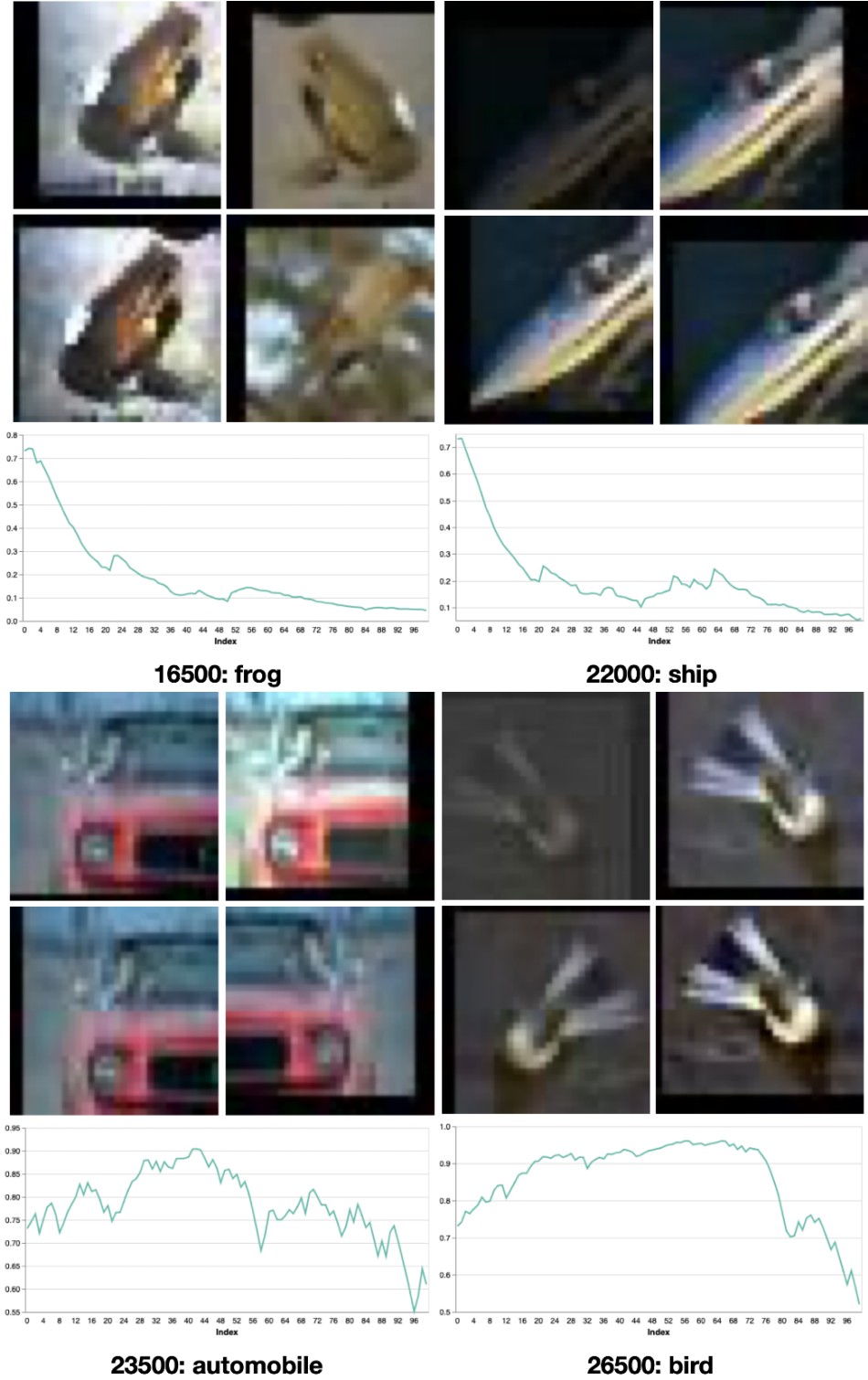

Figure 2: Additional visualizations of how augmentation instance parameters $(m_i)$ vary during training with AA augmentations for different sample types ("easy", "medium", and "hard") in CIFAR-10.

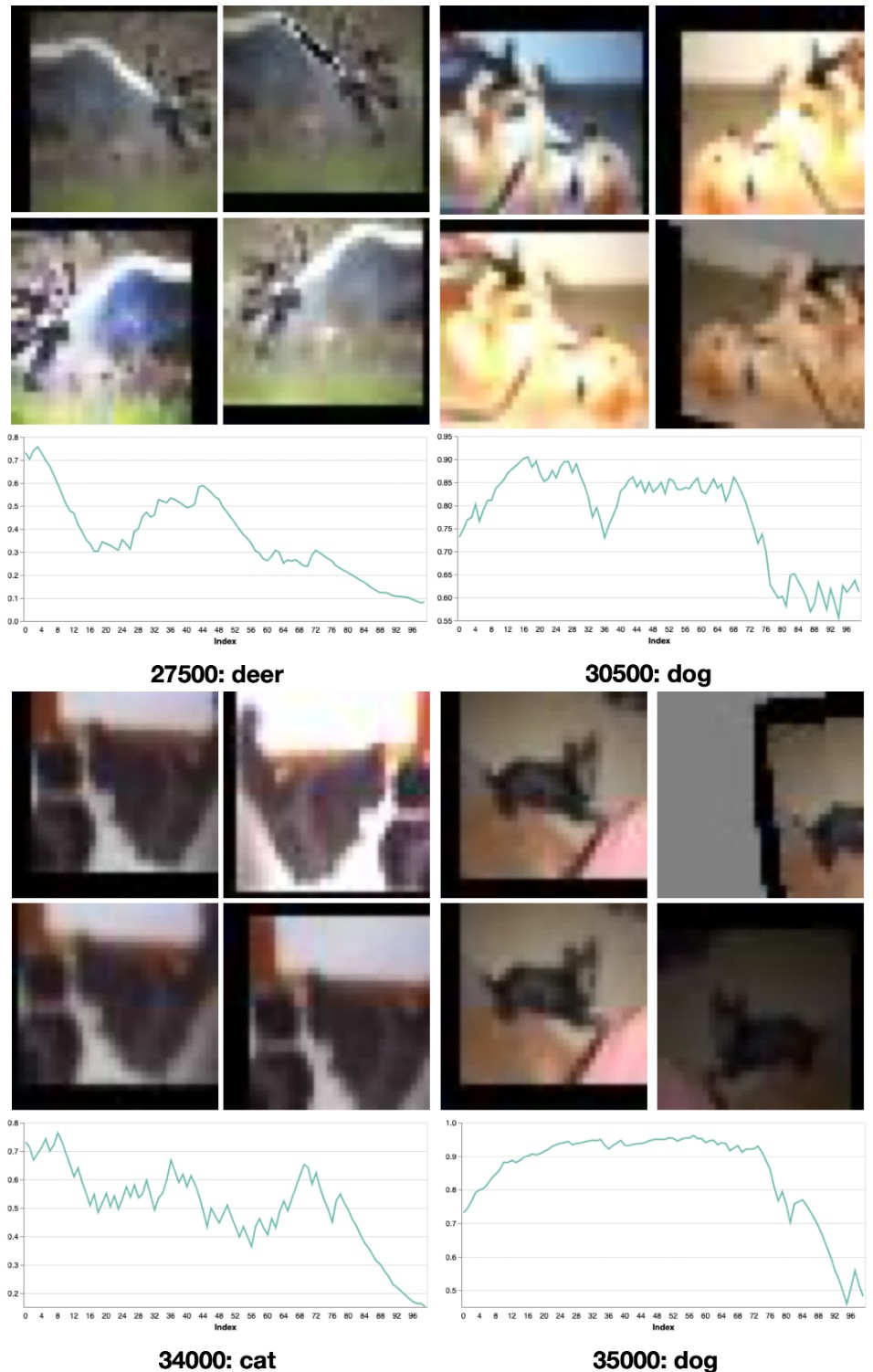

Figure 3: Additional visualizations of how augmentation instance parameters $(m_i)$ vary during training with AA augmentations for different sample types ("easy", "medium", and "hard") in CIFAR-10.

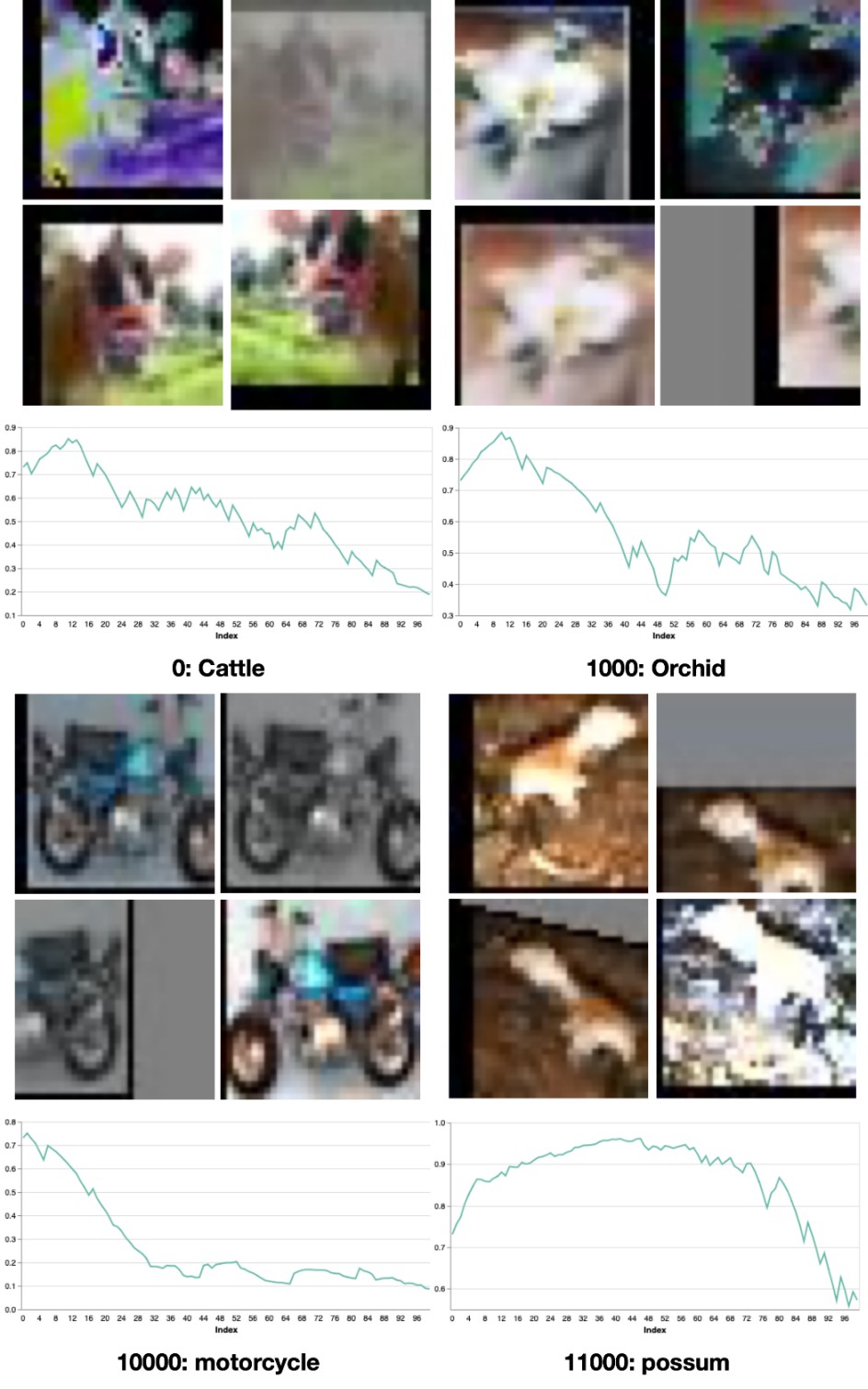

Figure 4: Additional visualizations of how augmentation instance parameters ($m_i$) vary during training with AA augmentations for different sample types ("easy", "medium", and "hard") in CIFAR-100.

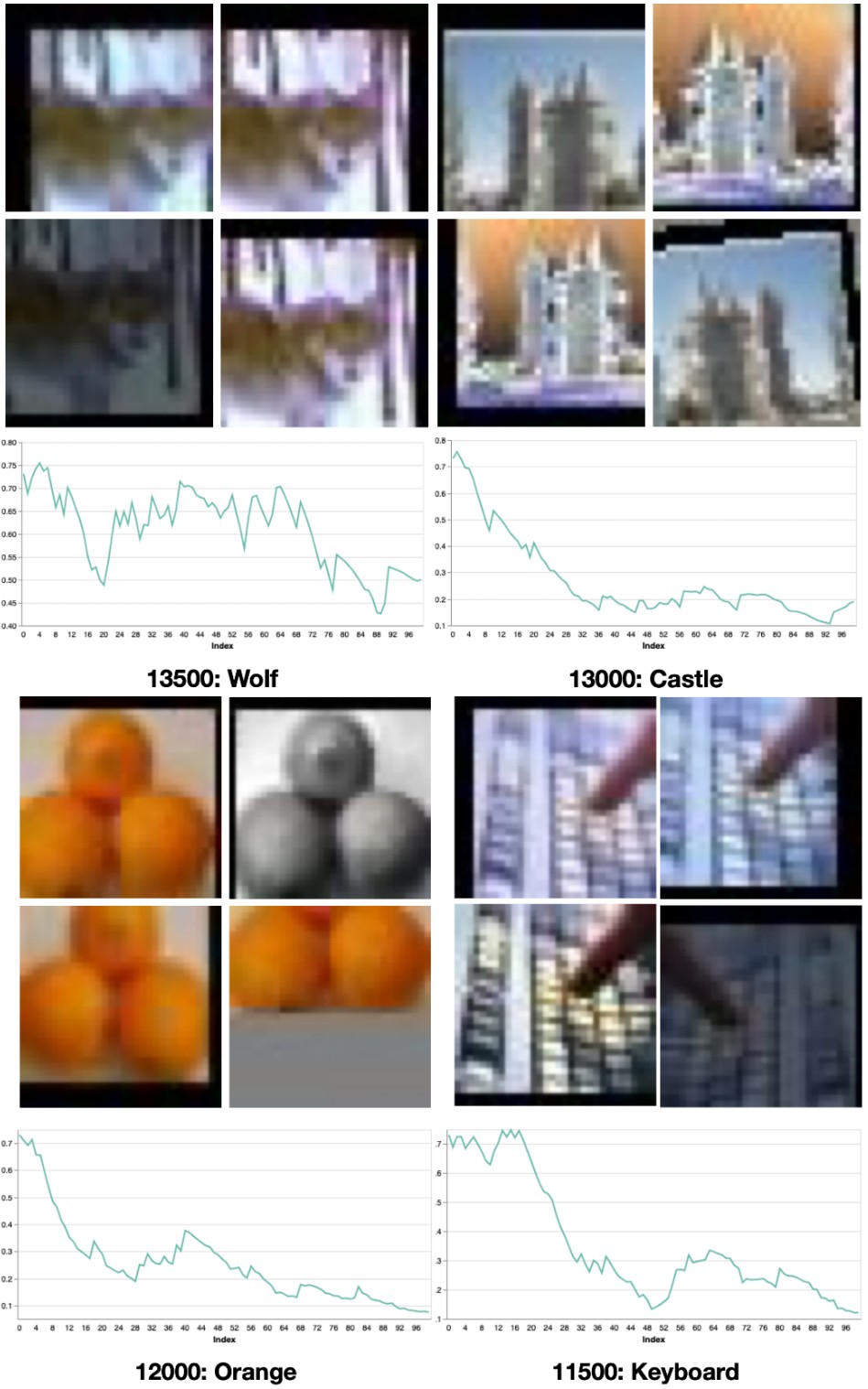

Figure 5: Additional visualizations of how augmentation instance parameters $(m_i)$ vary during training with AA augmentations for different sample types ("easy", "medium", and "hard") in CIFAR-100.

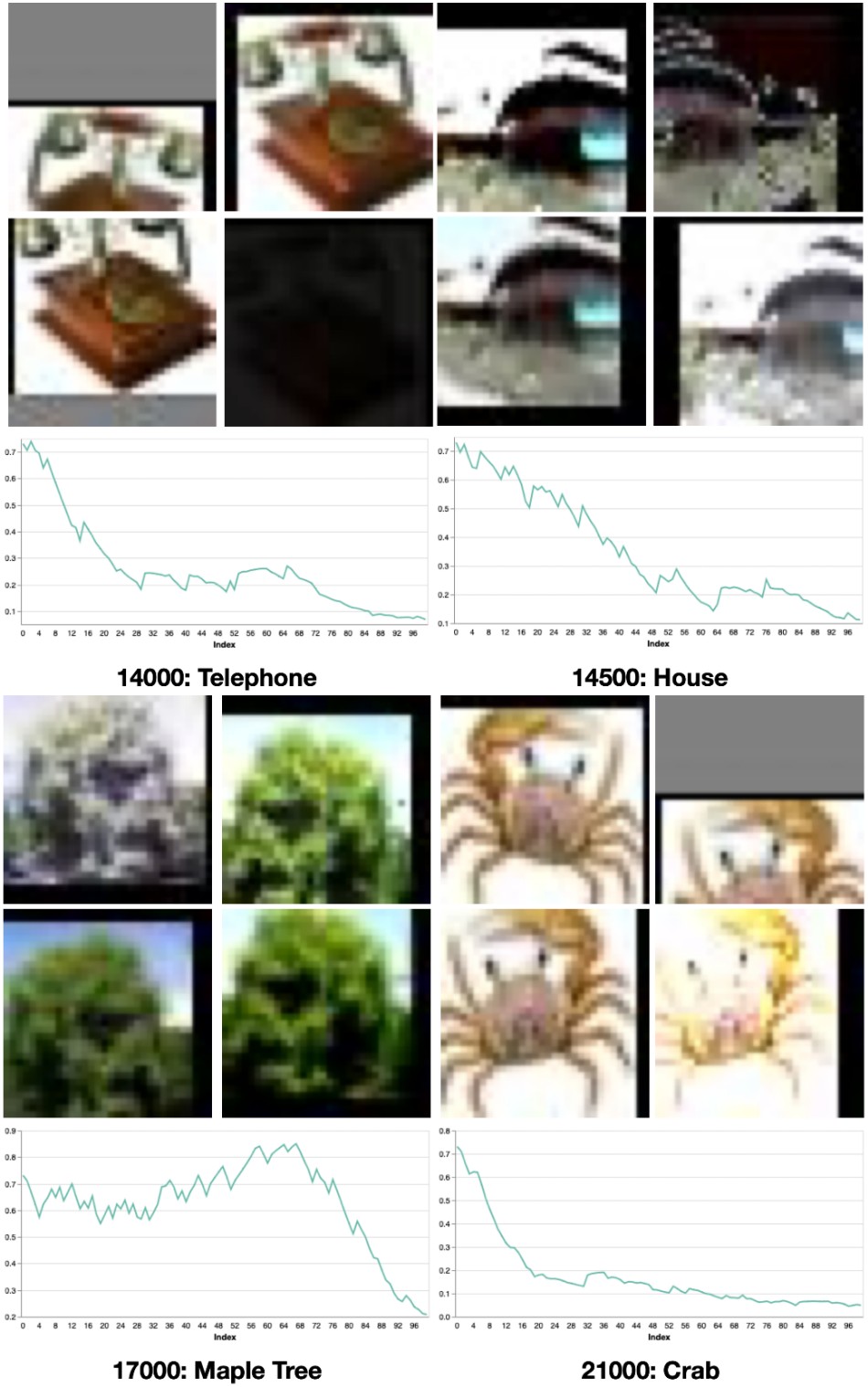

Figure 6: Additional visualizations of how augmentation instance parameters ($m_i$) vary during training with AA augmentations for different sample types ("easy", "medium", and "hard") in CIFAR-100.

# E  TRAINING STABILITY OF SPAUG

Because augmentations are inherently stochastic, the loss associated with a specific sample can fluctuate, introducing some level of noise in the selection of easy and hard samples. As a result, one might anticipate that SPAug could encounter some instabilities during training. However, Figure 7 illustrates the cross-entropy (CE) loss when training the WRN-40-2 backbone with the AugMix policy, both with and without SPAug. As shown, no training instabilities are observed.

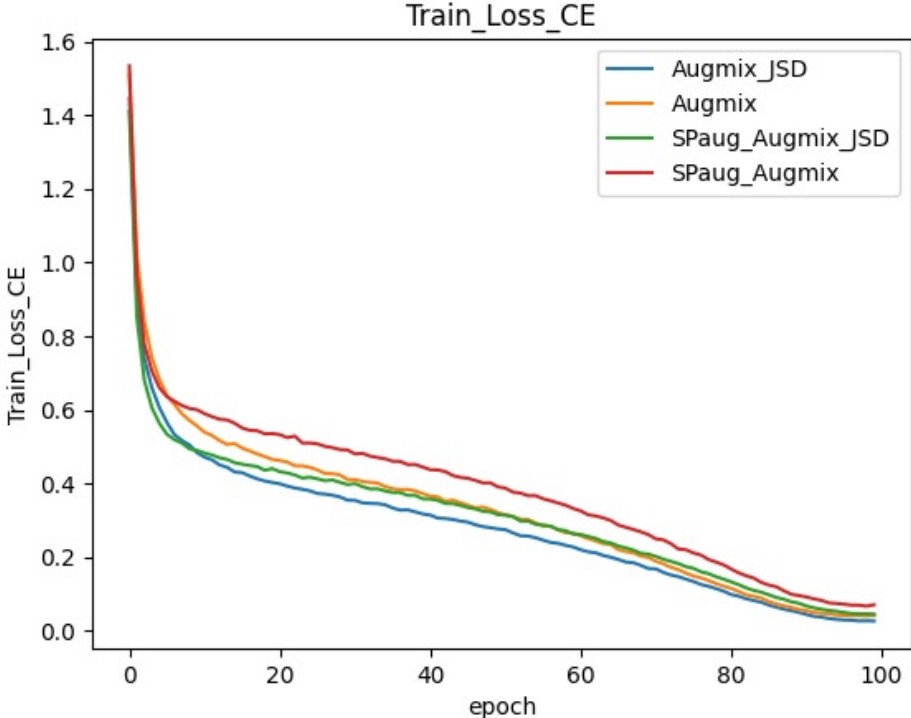

Figure 7: CE loss comparision w/ and w/o SPAug with AugMix training on CIFAR10 dataset with WRN-40-2 backbone.