# OpenReview forum: "Self-Paced Augmentations (SPAug) for Improving Model Robustness"
_ICLR.cc/2024/Conference — Submitted to ICLR 2024_

### Official Review · Reviewer_56By · 2023-10-30

**Soundness:** 3 good
**Presentation:** 3 good
**Contribution:** 3 good
**Rating:** 5
**Confidence:** 3

**Summary:**

The paper introduces an adaptive data augmentation strategy for deep neural networks, focusing on enhancing model robustness and performance. By employing a self-paced augmentation method, the research dynamically adjusts the intensity of data augmentation based on individual sample characteristics. The paper demonstrates the effectiveness of this approach using various datasets.

**Strengths:**

(1) The point of the paper is very good.
(2) Combining with multiple strategies to demonstrate performance improvement is also a great approach.

**Weaknesses:**

(1) In Section 4.3, you mentioned that the learnable SPAug has a significant improvement effect on the performance of AugMix in processing corrupted data. However, in the above Table 2, compared to the optimal model, your performance improvement is very limited or there is no improvement at all. The superiority of the model is not sufficiently reflected.
(2) You did not discuss the threshold τ in the subsequent experiments. It is not clear how you optimized the threshold. It feels like you are showing the best experimental results that you got separately with τ=0.1 or τ=0.2.
(3) The last two models, after adding SPAug-Learnable, indeed improved on the corrupted dataset, but there was a loss on the clean dataset. I hope you can consider the adaptability issue between the specific data augmentation strategy and the data.
(4) Introducing an adaptive learning data augmentation strategy increases the complexity of the model and may lead to extended training times and computational costs. While you mentioned that this method introduces minimal overhead, there is no specific experimental data in the article to support this claim.

**Questions:**

Please refer to the weaknesses.

---

> ### Author Response · Authors · 2023-11-23
> **Response to Reviewer 56By Q1**
>
> >  In Section 4.3, you mentioned that the learnable SPAug has a significant improvement effect on the performance of AugMix in processing corrupted data. However, in the above Table 2, compared to the optimal model, your performance improvement is very limited or there is no improvement at all. The superiority of the model is not sufficiently reflected.
>
> Thank you for your comment! After reviewing your feedback, we realized that the performance improvement of SPAug over AugMix may not have been adequately highlighted. The tables below provides a summary of the performance enhancements achieved by incorporating SPAug with the AugMix policy.
>
> | Dataset   | JSD? | AugMix          | AugMix+SPAug (Ours) | % Improvement |
> |-----------|------|-----------------|----------------------|--------------|
> | CIFAR-10  | No   | $14.0\pm{0.4}$  | $13.0\pm{0.2}$       | +1.0         |
> | CIFAR-10  | Yes  | $11.2\pm{0.3}$  | $11.0\pm{0.2}$       | +0.2         |
>
> Table 1: Comparison of SPAug+AugMix vs. AugMix in terms of mean corruption error on CIFAR-10-C with WRN-40-2 backbone.
>
> | Dataset     | JSD? | AugMix          | AugMix+SPAug (Ours) | % Improvement |
> |-------------|------|-----------------|----------------------|--------------|
> | CIFAR-100   | No   | $40.0\pm{0.1}$  | $39.0\pm{0.1}$       | +1.0         |
> | CIFAR-100   | Yes  | $36.1\pm{0.1}$  | $35.0\pm{0.1}$       | +1.1         |
>
> Table 2: Comparison of SPAug+AugMix vs. AugMix in terms of mean corruption error on CIFAR-100-C with WRN-40-2 backbone.
>
> | Dataset   | JSD? | AugMix          | AugMix+SPAug (Ours) | % Improvement |
> |-----------|------|-----------------|----------------------|--------------|
> | CIFAR-10  | No   | $10.7\pm{0.2}$  | $10.0\pm{0.1}$       | +0.7         |
> | CIFAR-10  | Yes  | $9.3\pm{0.0}$   | $8.7\pm{0.0}$        | +0.6         |
>
> Table 3: Comparison of SPAug+AugMix vs. AugMix in terms of mean corruption error on CIFAR-10-C with WRN-28-10 backbone.
>
> | Dataset     | JSD? | AugMix          | AugMix+SPAug (Ours) | % Improvement |
> |-------------|------|-----------------|----------------------|--------------|
> | CIFAR-100   | No   | $33.6\pm{0.3}$  | $33.3\pm{0.0}$       | +0.3         |
> | CIFAR-100   | Yes  | $31.9\pm{0.2}$  | $31.3\pm{0.2}$       | +0.6         |
>
> Table 4: Comparison of SPAug+AugMix vs. AugMix in terms of mean corruption error on CIFAR-100-C with WRN-28-10 backbone.
>
>
> As demonstrated in the tables above, it is evident that coupling SPAug with AugMix consistently results in a significant improvement in performance on both CIFAR-10 and CIFAR-100, whether using the WRN-40-2 or WRN-28-10 backbone. This further underscores the benefit of employing instance-specific augmentation parameters over uniform augmentation parameters.

---

> > ### Author Response · Authors · 2023-11-23
> > **Response to Reviewer 56By Q2**
> >
> > > You did not discuss the threshold $\tau$ in the subsequent experiments. It is not clear how you optimized the threshold. It feels like you are showing the best experimental results that you got separately with $\tau=0.1$ or $\tau=0.2$.
> >
> > Thank you for the comment.
> >
> > For each dataset and the backbone architecture, we performed a hyperparamter tuning with $\tau$ values of 0.0, 0.1, 0.25, 0.5, 0.75, 0.9, and 1.0. The presented results in the paper are for the optimal configuration of $\tau$.
> >
> > It was observed that, in most cases, a $\tau$ value of 0.75 or 0.9 yielded the best results. Also, SPAug improves over the baseline for a wide range of choices for $\tau$ demonstrating it's robustness to this hyper-parameter as it can be seen in Tables 3, and 4 in the supplementary material.

---

> > > ### Author Response · Authors · 2023-11-23
> > > **Response to Reviewer 56By Q3**
> > >
> > > > The last two models, after adding SPAug-Learnable, indeed improved on the corrupted dataset, but there was a loss on the clean dataset. I hope you can consider the adaptability issue between the specific data augmentation strategy and the data.
> > >
> > > Thank you for the comment. While we may not have fully understood the reviewer's concern, we will provide our response based on our interpretation. In many cases, when we introduce heavy augmentation during training to enhance performance on corrupted data (generalization to corrupted data), there can be a trade-off with the performance on clean data. Achieving the best performance on both clean and corrupted test data simultaneously can be challenging.
> > >
> > > However, based on the experimental results presented in our paper, we have observed that in most cases, SPAug achieves better clean and corrupted accuracy compared to its corresponding uniform augmentation policy. This improvement can be attributed to the fact that SPAug applies varying levels of augmentation intensities to each sample based on its "easiness" or "hardness." Consequently, easy samples undergo training with a maximum level of augmentation intensity, while harder samples receive a minimum level of augmentations. We believe that this balanced application of augmentation intensity contributes to the improvement in both corrupted and clean test accuracy.
> > >
> > > We hope this clarifies our approach and its impact on clean and corrupted data performance.

---

> ### Author Response · Authors · 2023-11-23
> **Response to Reviewer 56By Q4**
>
> > ntroducing an adaptive learning data augmentation strategy increases the complexity of the model and may lead to extended training times and computational costs. While you mentioned that this method introduces minimal overhead, there is no specific experimental data in the article to support this claim.
>
> That is a valid question, and thank you for highlighting it. Since our method does not involve an expensive optimization process (such as the bi-level optimization procedure utilized in AugMax), we did not observe any significant slowdown compared to the baseline. To support our argument, we benchmarked the training time per epoch for standard training, AugMix, SPAug with AugMix policy, and AugMax on ImageNet with ResNet18, using a single NVIDIA A6000 GPU.
>
> | Method                | Time (sec/epoch) |
> |-----------------------|-------------------|
> | Standard              | 2669              |
> | AugMix                | 3622              |
> | **SPAug (w/ AugMix)**     | **3698**              |
> | AugMax                | 5264              |
>
> As shown in the table above, introducing SPAug into a AugMix slows down the code only by 3698-3622  =  76 sec/epoch  which is very minimal computational overhead compared to augmentation policies that use complex optimization processes to determine instance-specific parameters, like AugMax, which can result in considerable computational overhead.

---

### Official Review · Reviewer_AiNq · 2023-10-30

**Soundness:** 2 fair
**Presentation:** 2 fair
**Contribution:** 2 fair
**Rating:** 3
**Confidence:** 4

**Summary:**

The author proposed self-paced augmentations for training neural networks. In particular, it chooses the data augmentation strength dynamically according to the training statistics of training samples. The author conducts experiments on CIFAR10/CIFAR100 and demonstrates its effectiveness and robustness.

**Strengths:**

i) The idea and the implementation are easy to follow, and the writing is clear to read.

ii) The author provides visualization and quantitative analysis to validate the effectiveness of the proposed method.

**Weaknesses:**

i) The experiments are only conducted on some small datasets (e.g. CIFAR), it is not convincing without the experiment results on a large-scale dataset(e.g. ImageNet)

ii) The proposed augmentation, as illustrated in Eq.1, shares a similar formulation as two widely used augmentation methods: CutMix and Mixup. However, there is no comparison between the proposed method and CutMix/Mixup

iii) There are also some related works[a,b], which also adjust the augmentation strength according to other training statistics. They are not discussed and compared in experiments.

[a] Universal Adaptive Data Augmentation
[b] ADAPTIVE DATA AUGMENTATION FOR IMAGE CLASSIFICATION

**Questions:**

Refer to the weakness

---

> ### Author Response · Authors · 2023-11-22
> **Response to Reviewer AiNq Q1 & Q2**
>
> > The proposed augmentation, as illustrated in Eq.1, shares a similar formulation as two widely used augmentation methods: CutMix and Mixup. However, there is no comparison between the proposed method and CutMix/Mixup.
>
> Thank you for the comment. Although the proposed approach may appear similar to CutMix and MixUp at first glance, they are fundamentally different from our SPAug. The primary distinction lies in the fact that CutMix and MixUp fall under a uniform augmentation policy. In other words, the hyperparameters for CutMix and MixUp are fixed for all samples in the dataset, resulting in uniform augmentation intensity applied to all samples. In contrast, SPAug introduces sample-dependent augmentation intensity during training, tailored to how each sample converges. Samples that converge rapidly (referred to as "easy samples") are encouraged to undergo higher levels of augmentation intensity, while those converging slowly (referred to as "hard samples") are encouraged to train with less synthetic augmentation applied. Moreover, instead of employing computationally expensive optimization processes such as reinforcement learning or bi-level optimization, we draw inspiration from the curriculum learning literature and propose to use each sample's training loss (cross-entropy loss) as a proxy measure to determine their ease or difficulty of convergence.
>
> However, in response to the reviewer's feedback, we have decided to compare our SPAug method with CutMix and MixUp to provide a more comprehensive analysis. Following your comment, we have conducted the comparison as shown in Table 1 and Table 2 (below).
>
> | Standard | CutOut | MixUp | CutMix | AA | AugMix | AugMix+SPAugLearnable |
> |----------|--------|-------|--------|----|--------|-----------------------|
> | 27.2     | 26.8   | 22.3  | 27.1   | 23.9 | 11.2   | 11.0                  |
>
> Table 1: Comparison of corrupted Test Error (C-Err.) of SPAug with AugMix policy on CIFAR-10-C with WRN-40-2.
>
>
>
> | Standard | CutOut | MixUp | CutMix | AA | AugMix | AugMix+SPAugLearnable |
> |----------|--------|-------|--------|----|--------|-----------------------|
> | 53.2     | 53.5   | 50.4  | 52.9   | 49.6 | 36.1   | 35.0                  |
>
> Table 2: Comparison of corrupted Test Error (C-Err.) of SPAug with AugMix policy on CIFAR-100-C with WRN-40-2.
>
> As shown in Table 1 and Table 2, the augmentation policies CutOut, MixUp, and CutMix result in significantly higher corrupted test error rates (C-Err.) than Augmix+SPAugLearnable. This indicates that models trained with those augmentation policies have lower generalizability against common corruptions.
>
> In regards to demonstrating results on ImageNet, we have done some preliminary comparisons in Table 3 of the main body. We will extend experiments on this dataset in future revisions.

---

> ### Author Response · Authors · 2023-11-22
> **Response to Reviewer AiNq Q3**
>
> Thank you for bringing our attention to these related works. However, we would like to clarify the differences between the proposed SPAug and two other related methods, Universal Adaptive Data Augmentation (UADA) [1] and Adaptive Data Augmentation [2].
>
> When comparing our work to UADA, the main distinction lies in the fact that, although UADA adjusts the augmentation parameters adaptively over iterations, it maintains uniform parameters for all data samples in the dataset. In other words, UADA's goal is to change (or adapt) the augmentation parameters based on the model's gradient information during training. In contrast, our objective is to have instance-dependent augmentation parameters that change based on how each sample converges during training. We will discuss UADA in the related work section, highlighting this distinction between SPAug and UADA. However, it's worth noting that UADA does not provide a comparison of their performance on corrupted test sets (such as CIFAR-10-C and CIFAR-100-C), which currently prevents us from making a direct comparison between UADA and SPAug.
>
> The main intuition behind the Adaptive Data Augmentation framework [2] is to find a small transformation that results in the maximum classification loss, leading to training with worst-case data augmentation. However, their proposed worst-case augmentation framework only works with translational augmentations and cannot be directly extended to more complex augmentation operations, such as color transformations or mixing. Furthermore, they have validated their results on very small datasets, such as MNIST-500 and Small-NORB, which prevents us from making a direct comparison with their work. Nevertheless, we will discuss this work in the Related Work section since it represents one of the early attempts to demonstrate the importance of adaptive data augmentations.

---

### Official Review · Reviewer_JB6n · 2023-11-03

**Soundness:** 3 good
**Presentation:** 2 fair
**Contribution:** 2 fair
**Rating:** 5
**Confidence:** 4

**Summary:**

This paper introduces the used of a self-paced algorithm for training with data augmentation. Each sample is now a linear combination of a sample without transformations plus a sample with augmentations, where the blending factor is learned during training with a self-paced strategy based on the loss. In practice, during training, easy samples with low loss are augmented more than hard samples with high loss, that are more difficult to learn. This approach is supposed to improve results, especially on corrupted datasets. Results are presented for CIFAR 10/100 original and corrupted.

**Strengths:**

- The idea of using self-paced learning for augmentations is new up to my knowledge and makes sense.
- The method can be adapted to many kind of data augmentation by adding a few lines of code as shown in Algorithm 1.

**Weaknesses:**

- The method is not compared directly with other reported results and the provided baselines seems to be weak, making results not accurate. For instance, in RA the error reported on CIFAR10/100 for Wide-ResNet 28-10 are respectively 2.7 and 16.7, while in the proposed paper are 3.3 and 19.1.
- Authors state that all previous data augmentation models use augmentations that are not instance specific. However, there are papers, (eg. [1] or [2]) that learn instance specific augmentation through a neural network. Authors should cite the family of data augmentation methods based on transformations learned by a network and if possible compare with them.
- Results are limited to CIFAR10/100. Results on a larger dataset as ImageNet should be provided.
- In related work, there should be a part considering self-paced methods. There is a vast literature on such kind of approaches and even if it is not applied to data augmentation it is still relevant. Some approaches are cited during the presentation of the method, but I think that a more exhaustive presentation in related work is needed.
- With large datasets, you need to store a large number of parameters, one per sample.

[1] Miao et al., "Learning Instance–Specific Augmentations by Capturing Local Invariances", ICML 2023

[2] Benton et al. "Learning invariances in neural networks", NeurIPS 2020

**Questions:**

- Due to the stochastic nature of the augmentations, the corresponding loss for a given sample can fluctuate and introduce a certain noise on the selection of the easy/hard samples. Did you find any instabilities in the training due to this noise?
- Why you train with a limited budget instead of training until convergence? This makes results not compareble with the state of the art.
- What is the reason to propose the toy experiment in section 4.2. The following evaluations are also performed on the same dataset.
- What is the value of the thresholds $\tau$ for the experiments in Table 2?

---

> ### Author Response · Authors · 2023-11-22
> **Response to Reviewer JB6n Q1**
>
> > Due to the stochastic nature of the augmentations, the corresponding loss for a given sample can fluctuate and introduce a certain noise on the selection of the easy/hard samples. Did you find any instabilities in the training due to this noise?
>
> Thank you for your comment. No, we did not observe any training instabilities during the training.
>
> For example, **Figure 7 in Supplementary Document** shows CE loss when training WRN-40-2 backbone with AugMix policy w/ and w/o SPAug. As can be seen, there is no training instabilities observed.

---

> > ### Author Response · Authors · 2023-11-22
> > **Response to Reviewer JB6n Q2**
> >
> > > Why you train with a limited budget instead of training until convergence? This makes results not comparable with the state of the art.
> >
> > Thank you for your comment! The reason we show results for both 100 epochs and 200 epochs (standard training epochs used in most previous works) is to demonstrate that when SPAug is included, the network tends to converge faster than with a uniform augmentation policy. As observed by another reviewer, uniform augmentation policies require more time than standard training to fully converge. For example, when considering the AA policy on CIFAR100 with a WRN-28-10 backbone, training for 200 epochs improves performance by +0.9\% (40.5\% to 39.6\%) compared to training for 100 epochs. However, if we incorporate SPAug into AA, the performance improves by +0.4\% (39.3\% to 38.9\%) because even with 100 epochs, we can see that the network has converged to a better optimum than with the uniform policy. As we emphasized in the introduction, our goal in incorporating instance-dependent augmentation parameters is primarily to enhance convergence, which is why we report results for both the standard training epochs and half of the standard training epochs.

---

> > > ### Author Response · Authors · 2023-11-22
> > > **Response to Reviewer JB6n Q3**
> > >
> > > > What is the reason to propose the toy experiment in section 4.2. The following evaluations are also performed on the same dataset.
> > >
> > > Thank you for the comment! The reason for the toy experiment in Section 4.2 is to demonstrate how SPAug improves performance with a hand-crafted augmentation policy. Typically, when we begin training a neural network, we have no prior knowledge of the best augmentation policy for a given task. For instance, we might start with very basic augmentations like random resized crop and flip, then gradually add color augmentations such as random intensity, hue, saturation, etc., and even incorporate perspective transformations. Since the optimal hyperparameters for these augmentations are often unknown or only partially determined through hyperparameter tuning, the toy experiment suggests that adding SPAug on top of such suboptimal augmentation parameters still leads to performance improvement. The remaining results presented in the paper aim to demonstrate that SPAug can even enhance performance compared to well-optimized augmentation policies previously established for each dataset.

---

> > > > ### Author Response · Authors · 2023-11-22
> > > > **Response to Reviewer JB6n Q3**
> > > >
> > > > > What is the value of the thresholds  for the experiments in Table 2?
> > > >
> > > > Thank you for the comment. The values reported in the paper for the AugMix + SPAug policy with WRN-40-2 and WRN-28-10 backbones use threshold values of 0.75 or 0.9. We conducted hyperparameter tuning for $\tau$ with values ranging from 0.1 to 1.0. It was observed that, in most cases, a $\tau$ value of 0.75 or 0.9 yielded the best results. Also, SPAug improves over the baseline for a wide range of choices for $\tau$ demonstrating it's robustness to this hyper-parameter as it can be seen in Tables 3, and 4 in the supplementary material.

---

> > > > > ### Author Response · Authors · 2023-11-22
> > > > > **Response to Reviewer JB6n**
> > > > >
> > > > > > RA numbers reported in the paper are not similar to those reported in the original paper.
> > > > >
> > > > > We thank you for raising this point. We should clarify that the difference lies in the fact that the authors of the RA paper, apply cutout after RandAugment in the results they report. In our experiments, since our goal was to directly compare with RandAugment, we did not apply cutout to the baseline (i.e., RA) and to SPAug+RandAug. We will make this clear in the next revision and also include the comparison to RandAugment + Cutout as well.

---

### Official Review · Reviewer_6RhX · 2023-11-04

**Soundness:** 3 good
**Presentation:** 2 fair
**Contribution:** 2 fair
**Rating:** 5
**Confidence:** 4

**Summary:**

The authors propose an adaptive augmentation technique which dynamically adjust the augmentation intensity based on training statistics. The approach can be applied to existing augmentation framework, such as AugMix, RandomAugment and AutoAugment. The experimental results show that it can improve model robustness to image corruptions.

**Strengths:**

+ Interesting idea to combine curriculum learning into data augmentation, controlling how models learn from augmented samples.
+ The writing is well-structured and easy to read.

**Weaknesses:**

__Missing comparisons with existing work__
The authors did not explain the difference of their approach from AugMax [1] framework, which combines augmented images with adversarially calculated weights.  Similarly, Hou et al. [2] adopted the idea of curriculum learning and applied it to decide when to augment data during training. But this is also not discussed by the authors. Other augmentation techniques being sota, PRIME [4] and TrivialAugment [5], are not compared with.

__clarity__
Explanation of equation (4) is not clear and in Fig. 3, the formula for hard sample should be: L_i - \sigma(m_i)
It is unclear in the formula whether the cross-entropy loss should be calculated on the original images or the augmented images. If the cross-entropy loss is computed based on the augmented images, then the parameter m_i could be also updated through backpropagation, just like AugMax [1].

__Non-comprehensive experimental results__
The authors only show results of SPAug combined with AugMix on ImageNet, while the results of it combined with AutoAugment and RandomAugment are not provided. The standard performance of models are not given in Table 2. The evaluation metrics in [3] used for benchmarking the robustness of models to image corruptions are not used.
Experiments are mostly focused to small datasets, while ImageNet is only used for a single comparison. The claims and observations made on CIFAR10/100 are thus limited and cannot be generalized and compared with those in other papers that extensively experiments on ImageNet. Furthermore, only a single architecture is tested: how would this method perform with transformer training strategies?

__Figures do not have enough explanations__
The meaning of x and y axes in Fig, 4 are not explained and  the figure itself is not easily readable. From the figure, I cannot interpret how the binary mapping function governs the extent of augmentations. In Fig. 5, the authors provide four augmented versions for one class, alongside the changes of m_i during training. However, the relationship between the augmented images and the m_i tendency is not explained.

__Missing appendices__
The authors mention appendix and supplementary materials, but it is not given.

[1] Wang et al., “AugMax: Adversarial Composition of Random Augmentations for Robust Training”, (2021)
[2] Hou et al., “When to Learn What: Model-Adaptive Data Augmentation Curriculum”, (2023)
[3] Hendrycks et al., “Benchmarking Neural Network Robustness to Common Corruptions and Perturbations”, (2019)
[4] Modas et al., PRIME: A few primitives can boost robustness to common corruptions, (2022)
[5] Muller et al. TrivialAugment: Tuning-free Yet State-of-the-Art Data Augmentation, (2021)

Minor: Implementation python code directly pasted as pseudo code.

**Questions:**

- How is the work different from [1,2] and what improvements have been made regarding them? This work is quite similar to AugMax[1] in the sense of combining augmented images with their original version using weights calculated by backpropagation.
- In the experiments, how is the threshold τ that distinguishes easy samples from hard ones in the minibatch determined? Is it affected by the minibatch size? Is there any trade-off between them, considering needed computational resources?
- In Tables 4 and 5, the results of models trained in the 100 and 200 epochs are given. However, the gained performance through more training epochs is not significant. For instance, in Table 4, the gained C-Err for SPAug-Learnable is on average 0.5, while this value for AA is 0.9. What is the reason for training more epochs to obtain trivial improvement using SPAug-Learnable? Besides, why is the baseline model WRN-28-10 having different C-Err in Table 2 and 4?
- How does the method perform on ImageNet with other existing augmentation techniques, and with other architectures (e.g. Transformers)?

---

> ### Author Response · Authors · 2023-11-22
> **Response to Reviewer 6RhX Q1**
>
> We thank the reviewer for pointing out these interesting related works [1-2]. While we recognize that the high-level idea of AugMax, MADAug, and SPAug is to have instance specific augmentation parameters, our approach of addressing this problem fundamentally differs from others.
>
> *Answer to Q1*
> > How is the work different from [1,2] and what improvements have been made regarding them? This work is quite similar to AugMax[1] in the sense of combining augmented images with their original version using weights calculated by back-propagation.
>
> When considering AugMax alongside SPAug, we would like to emphasize the following differences:
> - In AugMax, these instance-specific parameters are determined in an adversarial manner. Hence, during training, each sample is trained with its \textbf{maximum} augmentation level. In contrast, SPAug determines these parameters based on each sample's Cross-Entropy (CE) loss at the previous epoch.
> - Since AugMax determines the instance-specific parameters in an adversarial manner, for each sample, at each training iteration, they need to perform a potentially expensive adversarial attack, which adds computational overhead. In contrast, SPAug uses the sample's previous epoch loss as the proxy value, hence there is little to no computational overhead over standard training.
> - In addition to the above two main differences, AugMax introduces changes to the network architecture to boost its performance. Since, in AugMax, samples are trained with their maximum augmentations, it leads to a deviation of distribution between augmented and input data, which makes the training more challenging, as noted in the paper. To address this issue, they use a different normalization technique termed DuBIN, to disentangle the instance-wise feature heterogeneity of AugMax samples. Since SPAug controls augmentation intensity based on its previous loss values, the augmented views won't deviate much from the original training distribution as in AugMax. In contrast, our method (SPAug) does not require additional modifications to network architecture or the training setup and can be easily integrated to existing training pipelines. When we use the same basic WRN40-2 architecture (with regular 2D batchnorm) that we use in our paper, we find that on CIFAR-100 AugMax has 36.6\% corrupted error (which is 1.6\% worst than that of SPAug) . In the next revision, we will include this comparison.
>
> When considering the second work (MADAug), we want to draw the reviewer's attention to the fact that it was posted on arXiv on Sat, 9 Sep 2023, which is very close to the ICLR submission date and can be considered as a concurrent related work. Similar to SPAug, MADAug also proposes having instance-wise augmentation parameters, further demonstrating the recent trend in this direction. However, after carefully reviewing their approach, we have observed the following fundamental differences with our work:
> - They also acknowledge that at the beginning of training, samples are not well-fitted; hence applying the same level of augmentations at that phase reduces the final accuracy. Therefore, they propose to use a manually designed probability schedule $p(t) = \tanh(t/\tau)$, where $t$ denotes the current training epoch. In addition, MADAug assigns an augmentation probability $p$ and magnitude $\lambda$ to each sample, determined by the policy network parameterized by $\theta$, which takes the image features extracted from the task model parameterized by $w$. The bi-level optimization process consists of three steps:
>    1. One-step gradient descent on $w_t$ to achieve a closed-form surrogate $\hat{w}$ of the lower-level problem solution.
>    2. Updating policy network parameters $\theta_t$ by minimizing the validation loss computed by the meta-task model $\hat{w}t$ on a mini-batch of the validation set.
>    3. Updating the main model parameters $w$ based on the parameter $\theta_{t+1}$ of the policy model.
>
> Due to this bi-level optimization, MADAug requires 2 forward passes for each training minibatch as well as forward passes on the validation minibatch, hence taking approximately $1.5\times$ longer training time per epoch compared to our method when training Wide-ResNet-28-10 on the CIFAR-10 dataset.

---

> ### Author Response · Authors · 2023-11-22
> **Response to Reviewer 6RhX Q2**
>
> > In the experiments, how is the threshold $\tau$ that distinguishes easy samples from hard ones in the minibatch determined? Is it affected by the minibatch size? Is there any trade-off between them, considering needed computational resources?
>
> We thank the reviewer for this interesting question. As noted in the supplimantary material, the threshold $\tau$ is determined like regular hyperparamter tuning. We find the best threshold $\tau$ for a given dataset by performing hyperparamter seach over $\tau$ values ranging from $0.0, 0.1, 0.25, 0.5, 0.75, 0.9, 1.0$. We did not change the batch-size values throughout the experiments and keep it same as the same batch size used in the standard training.
>
> For example, Table 3 and 4 in **Supplementary Document** shows how the SPAug results varying for hyperparamter tuning on $\tau$ for CIFAR-10 and CIFAR-100 datasets with AugMix. As shown in the tables, we observed that $\tau = 0.75$ and $\tau = 0.9$ works well and can be considered as a good rule of thumb.

---

> ### Author Response · Authors · 2023-11-22
> **Response to Reviewer 6RhX Q3**
>
> > In Tables 4 and 5, the results of models trained in the 100 and 200 epochs are given. However, the gained performance through more training epochs is not significant. For instance, in Table 4, the gained C-Err for SPAug-Learnable is on average 0.5, while this value for AA is 0.9. What is the reason for training more epochs to obtain trivial improvement using SPAug-Learnable? Besides, why is the baseline model WRN-28-10 having different C-Err in Table 2 and 4?
>
> Thank you for the comment! The reason why we presented the results for 100 and 200 epochs is that we wanted to show that with SPAug, the network tends to converge pretty fast compared to having a uniform augmentation policy, as motivated in the introduction. You are absolutely correct in your assessment that with an additional 100 epochs of training, AA improves by +0.9\%, whereas with SPAug, it improved by +0.5\% because, with 100 epochs, it already converged to a better accuracy than AA. Another point we want to emphasize here is that usually when we incorporate an augmentation policy, it requires longer training time (like 200 epochs or more) to converge properly because with the uniform augmentation policy, some samples become much harder to fit, requiring longer training time. However, when we have instance-wise augmentation parameters, not all the samples undergo the same level of augmentation intensity, hence making the convergence easier and faster. We would like to thank the reviewer for this observation!
>
> Regarding your second point about why the baseline numbers in Table 2 and 4 are different, it's due to the stochastic nature of the experiments, and these are average values of two independent sets of 3 trials conducted for AugMix and AA. As we can see, each average accuracy has a non-zero standard deviation, hence the average value can fluctuate slightly. We can see that both average C-Err. values are within their standard deviation (for example, on CIFAR-10 with WRN-28-10 backbone: $23.5\pm0.9$ vs. $23.3\pm0.3$).

---

> ### Author Response · Authors · 2023-11-22
> **Response to Reviewer 6RhX Q4**
>
> > How does the method perform on ImageNet with other existing augmentation techniques, and with other architectures (e.g. Transformers)?
>
> Thank you for the comment. Following the previous works, we limited our backbones to ResNets and Wide-ResNet (WRN) architectures in the main body. We also included AllConvNet, DenseNet architectures as well in Table 2 of the **supplementary material**.) Similarly, following some of the literature, we have included ImageNet results in Table 3 on the ResNet-50 architecture.  However, we see the value of your comment, and we will try compare the results of SPAug with ViT architectures on ImageNet in the next revision, as it requires longer training time and some time to set up the experiments.

---

### Meta-Review · Area_Chair_Vd8d · 2023-12-18

**Metareview:**

This paper proposes a data-instance dependent augmentation parameterization, dynamically adjusting the intensity of augmentation in a self-paced manner based on the training loss statistics. Results are shown one several datasets including a toy datasets and CIFAR/SVHN and some limited experiments on ImageNet.

 While all of the reviewers appreciated that the idea itself makes sense, they had a range of issues related to the execution of the work and clarity of the paper, including 1) Lack of thorough comparison to the range of works touching this field (including similar ideas of instance-dependent parameters) as well as the use of very simple baselines under limited conditions (e.g. limited epochs) , 2) Lack of comparison on more significant datasets such as ImageNet, which are very limited in this paper, 3) Lack of clarity and a number of issues in terms of polish. While the authors did provide a rebuttal, they ignored a number of issues (e.g. they did not respond holistically to the range of issues raised by 6RhX and only answered some of the questions) and overall the reviewers agreed that the paper is not thorough or polished enough for acceptance.

**Justification For Why Not Higher Score:**

There are a number of unaddressed concerns by the reviewers.

**Justification For Why Not Lower Score:**

N/A

---

### Decision · Program_Chairs · 2024-01-16

Reject